# Rethinking Intracranial Aneurysm Vessel Segmentation: A Perspective from Computational Fluid Dynamics Applications

## Abstract

The precise segmentation of intracranial aneurysms and their parent vessels (IA-Vessel) is a critical step for hemodynamic analyses, which mainly depends on computational fluid dynamics (CFD). However, current segmentation methods predominantly focus on image-based evaluation metrics, often neglecting their practical effectiveness in subsequent CFD applications. To address this deficiency, we present the **I**ntracranial **A**neurysm **V**essel **S**egmentation (IAVS) dataset, the first comprehensive, multi-center collection comprising 641 3D MRA images with 587 annotations of aneurysms and IA-Vessels. In addition to image-mask pairs, IAVS dataset includes detailed hemodynamic analysis outcomes, addressing the limitations of existing datasets that neglect topological integrity and CFD applicability. To facilitate the development and evaluation of clinically relevant techniques, we construct two evaluation benchmarks including global localization of aneurysms (Stage I) and fine-grained segmentation of IA-Vessel (Stage II) and develop a simple and effective two-stage framework, which can be used as a out-of-the-box method and strong baseline. For comprehensive evaluation of applicability of segmentation results, we establish a standardized CFD applicability evaluation system that enables the automated and consistent conversion of segmentation masks into CFD models, offering an applicability-focused assessment of segmentation outcomes. The data, code, and model will be made publicly available upon acceptance.

## 1 Introduction

Intracranial aneurysm (IA) is a pathological dilation of blood vessels, mainly occurring at the branches and bifurcations of arteries (Schievink, 1997). IA is usually small and initially asymptomatic, but may gradually enlarge over time and lead to symptomatic manifestations, and even rupture in severe cases, resulting in a high incidence of morbidity and mortality (Cebral et al., 2005). Accurate assessment of rupture risk of IA is essential for medical intervention of neurovascular diseases (Etminan & Rinkel, 2016). Computational Fluid Dynamics (CFD) provides key biomechanical evidence for clinical decision-making by quantifying hemodynamic parameters such as wall shear stress and pressure distribution, which have been widely applied in various biomedical researches (Li et al., 2025; Morris et al., 2016; Wang et al., 2025).

Magnetic resonance angiography (MRA) serves as a non-invasive, high-resolution imaging modality that facilitates the detailed visualization of aneurysms, enabling the identification of their anatomical characteristics, including location, size, and complex morphological features (Pierot et al., 2013). Accurate segmentation of intracranial aneurysm and parent vessels (IA-Vessel) from MRA is an important step for subsequent CFD analysis (Patel et al., 2023). As manual localization and delineation remain a labor-intensive and time-consuming procedure for radiologists (Jiao et al., 2023), it is highly desirable to develop automated segmentation methods in clinical applications. With the unprecedented developments of deep learning, state-of-the-art segmentation methods have achieved comparable results with inter-rater variability (Isensee et al., 2021). As deep learning-based methods require labeled data for training, high quality open-source datasets have become a crucial foundation for the development of segmentation algorithms for various modalities of medical imaging (Antonelli et al., 2022; Gatidis et al., 2022; Ji et al., 2022; Ma et al., 2022; Qu et al., 2023).

Table 1: Summary of existing 3D MRA datasets for intracranial aneurysm segmentation tasks.

| Dataset | Volumes | IAs | IA-Vessel Mask | STL | IA-Vessel Centerline | Mesh | CFD Results |
|---------|---------|-----|----------------|-----|---------------------|------|-------------|
| ADAM | 113 | 156 | ✗ | ✗ | ✗ | ✗ | ✗ |
| INSTED | 191 | 68 | ✗ | ✗ | ✗ | ✗ | ✗ |
| Royal | 63 | 85 | ✓ | ✓ | ✗ | ✗ | ✗ |
| IAVS(Ours) | **641** | **587** | ✓ | ✓ | ✓ | ✓ | ✓ |

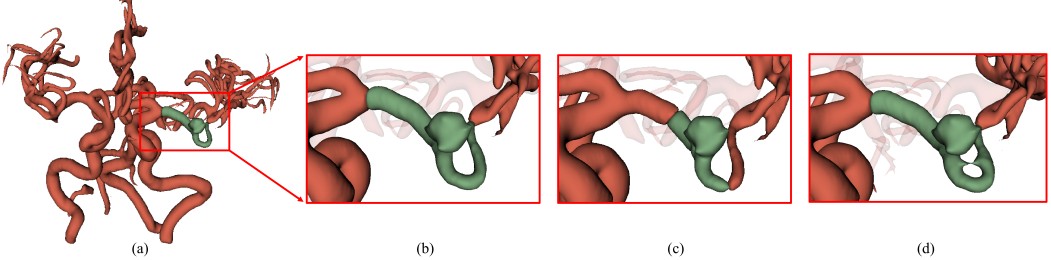

Figure 1: (a) Whole intracranial vasculature and local parent vessels. (b) IA-Vessel ground truth. (c) Despite the Dice score is relatively low (0.7648), no topological errors are present. (d) Although the Dice similarity coefficient is high (0.9869), topological errors are present which is unusable for CFD.

Despite the existence of several datasets for intracranial aneurysm segmentation, challenges persist when applying these datasets to hemodynamic analysis. First, there are structural deficiencies in the annotations of these datasets. Existing public datasets, such as ADAM (Timmins et al., 2021) and Royal(de Nys et al., 2024), generally lack refined annotations of the parent vessels and geometric validation labels. Additionally, they do not include records of hemodynamic results, which makes it challenging to support the end-to-end analysis process from image segmentation to CFD modeling. Second, the evaluation of segmentation results is limited. Most existing medical image segmentation models are assessed using region overlap-based metrics, such as the Dice coefficient. However, these metrics are insensitive to geometric topological abnormalities, including vessel adhesion and surface irregularities. These abnormalities usually fail CFD validation because of issues such as mesh generation failure or flow field distortion. Moreover, insufficient localization accuracy for small-sized aneurysms and the limited capability to maintain vascular connectivity further exacerbate the challenges in transitioning from image segmentation to biomechanical modeling.

To address these challenges, this study presents a systematic solution for segmenting intracranial aneurysms and vessels applicable to CFD, innovating across three sub-tasks: **dataset construction**, **benchmark design**, and **evaluation system**. The main contributions are outlined as follows.

- We collect and curate a large-scale multi-centre **I**ntracranial **A**neurysm **V**essel **S**egmentation (IAVS) dataset, comprising 641 3D MRA images and 587 annotations of aneurysms and IA-Vessels, including CFD analysis results. This dataset addresses the limitations of previous datasets that lack topological integrity and CFD applicability.

- We establish an standardised CFD applicability evaluation system that enables standardized estimation of CFD success probability given segmentation results. Additionally, we introduce a novel evaluation metric, the CFD-Applicability Score (CFD-AS), to facilitate a more comprehensive assessment of segmentation results.

- We conduct two evaluation benchmarks including global localization of aneurysms (Stage I) and fine-grained segmentation of IA-Vessel (Stage II) and develop a two-stage framework as a strong baseline for the accurate detection and segmentation of IA-Vessel, which significantly reduces geometric errors in segmentation masks and enhances CFD usability.

## 2 RELATED WORK

**Intracranial Aneurysm Datasets.** To accelerate the development of deep learning-based aneurysm and vessel segmentation, several segmentation datasets are evolved. However, existing public intracranial aneurysm datasets exhibit substantial limitations when applied to CFD studies. Regarding annotation completeness, the ADAM (Timmins et al., 2021) and INSTED (Chen et al., 2024) datasets offer 3D MRA images with aneurysm masks. However, they lack annotations of the parent vessels, which are essential for constructing CFD models. Conversely, the AneuX (Juchler et al., 2022) project provides preprocessed STL models for CFD but omits the original medical images and segmentation masks. In terms of anatomical accuracy, the Royal (de Nys et al., 2024) dataset includes both aneurysm outlines and vessel annotations. Nevertheless, several samples feature vessel adhesion, which undermine the validity of CFD boundary conditions. Similarly, the COSTA dataset (Mou et al., 2024) contains whole-brain vessel annotations, but suffers from adhesion errors in numerous distal branches of vessels, which inaccuracies directly impede the precision of CFD simulations. Overall, these works fail to provide a comprehensive database from image segmentation to CFD analysis, which underscores the necessity of developing application-oriented segmentation dataset.

**Aneurysm Vessel Segmentation.** Before the advent of deep learning, aneurysm and vascular segmentation relied mainly on classical vesselness-based methods, most notably the multiscale Hessian filter Frangi et al. (1998), with later benchmarks Lamy et al. (2022) revealing their variability across anatomical regions. However, these approaches struggle with complex aneurysmparent-vessel configurations and lack the geometric fidelity required for downstream hemodynamic analysis. Deep learning methods have shown excellent performance on several medical image segmentation tasks, yet aneurysm vessel segmentation presents unique challenges. Mainstream segmentation networks like 3D UNet (Çiçek et al., 2016) and nnUNet (Isensee et al., 2021) prioritize global voxel-wise accuracy but lack mechanisms for reliable small-target detection, essential for accurately segmenting both small aneurysms and fine vessels. Glia-Net (Bo et al., 2021) enhances aneurysm delineation via global context fusion but does not extend to parent-vessel segmentation. Object detection frameworks such as nnDetection (Baumgartner et al., 2021) achieve robust 3D lesion localization but falter on sub-voxel scale targets. Sphere-based detectors like CPM-Net (Song et al., 2020) and SCPM-Net (Luo et al., 2022) help stabilize small-object training dynamics but remain untested on vascular structures. Keypoint detection methods like MedLSAM (Lei et al., 2025) demonstrate promise for anatomical localization but have not been adapted for variable-size aneurysm center points. While AA-Seg (Yao et al., 2024a) pioneers joint aneurysmvessel segmentation, it still permits vessel adhesion across the aneurysm neck, highlighting the ongoing need for methods that can accurately and jointly segment both structures while respecting anatomical boundaries.

**Evaluation Metrics**. Conventional segmentation metrics inadequately capture the requirements of downstream CFD analysis. The Dice similarity coefficient (DSC) quantifies volumetric overlap but is insensitive to topological errors such as spurious vessel connections. Boundary IoU Cheng et al. (2021) improves edge accuracy assessment yet remains blind to global connectivity flaws. Centerline-aware metrics (clDice) Shit et al. (2021) incorporate explicit topological constraints but do not directly reflect mesh-generation feasibility or flow-convergence behavior. While innovative research into differentiable CFD solvers Yao et al. (2024b) aims to integrate physical simulations directly into the training loop, these methods are not yet directly applicable to the clinical task of intracranial aneurysm analysis due to the complex geometries and the non-differentiable nature of the traditional high-fidelity meshing and simulation pipeline required for clinical validation.

## 3 IAVS DATASET

**Motivation and Details.** Existing intracranial aneurysm datasets have structural deficiencies in the annotation and lack applicability in CFD applications. To bridge this gap, our IAVS dataset contains 641 3D MRA images and 587 aneurysms and IA-Vessels annotations with CFD analysis results, which is adapted from three existing datasets including ADAM (Timmins et al., 2021), INSTED (Chen et al., 2024) and Royal (de Nys et al., 2024), and a new in-house dataset from [hidden for review]. An overview of IAVS dataset is shown in Figure 2. For public datasets, the original ADAM and INSTED datasets only provide annotations of IAs. Despite the Royal dataset contains IA-Vessel mask and STL models, several samples feature vessel adhesion and are not applicable for CFD

Table 2: Statistics of IAVS dataset including data source, number and diameter of IAs.

| Dataset | No. of Images | | | No. of IAs per case | | | | IAs | Diameter of IA | | |
|---|---|---|---|---|---|---|---|---|---|---|---|
| | Total | Public | Private | 0 | 1 | 2 | ≥3 | | <3mm | 3-7mm | >7mm |
| Train | 467 | 175 | 292 | 82 | 345 | 34 | 6 | 432 | 55 | 272 | 105 |
| Set A | 76 | 76 | 0 | 42 | 29 | 3 | 2 | 41 | 16 | 17 | 8 |
| Set B | 98 | 0 | 98 | 0 | 85 | 10 | 3 | 114 | 10 | 93 | 11 |

Table 3: Statistics of imaging parameters in the IAVS dataset.

| Dataset Statistics | Min | Median | Max |
|---|---|---|---|
| Spacing (mm) | (0.21, 0.21, 0.30) | (0.36, 0.36, 0.50) | (0.47, 0.47, 1.20) |
| Volume Size (voxels) | (348, 384, 44) | (512, 512, 148) | (1024, 1024, 368) |

analysis. In contrast, our dataset contains CFD applicable segmentation masks and CFD analysis results, including 3D MRA images (1), voxel-level segmentation masks (2)-(3), geometric models (4)-(6) and CFD analysis results (7).

**Data Statistics.** The IAVS dataset is partitioned meticulously to meet the practical requirements of clinical research. Statistics of the proposed IAVS dataset in Table 2 reveals that the distribution of aneurysm quantity and size across cases closely mirrors clinical epidemiological patterns. This congruence effectively guarantees the representativeness of the dataset, enhancing the generalizability of the research findings. Additionally, all data underwent strict anonymization procedures and were rigorously reviewed and approved by the hospital ethics committee, ensuring full compliance with ethical standards. We split the images into 467 cases for training and validation, 76 cases from public datasets as Set A for internal evaluation, and 98 cases from in-house dataset as Set B for evaluation in clinical scenarios. For the training of Stage I, 373 cases are used for training and 94 cases are used for validation. In Stage II, candidate patches cropped based on IA annotation are used for training of IA-Vessel segmentation network. Following the same split of MRA images in Stage I, 357 patches are used for training and 99 patches are used for validation.

**Annotation.** After integrating medical imaging resources from three public and private datasets, we conduct annotations of IA and IA-Vessels for CFD applicable segmentation. The annotation workflow can be observed in Figure 2. IA annotations from the existing datasets are used if available. For in-house dataset, the annotations are completed and checked by experienced radiologists. For annotation of parent vessels of IA, we first use a pre-trained model using COSTA (de Nys et al., 2024) to preliminary segment whole-brain vessels of all MRA images. The pre-trained coarse-vessel model achieves Dice of 0.9204 on the official COSTA test set. Subsequently, focusing on the aneurysm-related vessel regions, the parent vessels are cropped and refined from the coarse segmentation of whole-brain vessels using 3D Slicer. The model-generated vessel segmentation mask is refined and verified by one CFD specialist and one board-certified radiologist instead of directly used without human correction. The refinement process strictly adheres to clinical anatomical principles, including eliminating abnormal geometric features and implementing an adaptive truncation strategy based on vascular bifurcation topology. When the parent vessel extends to the bifurcation, if the length of this segment of the vessel exceeds its diameter, truncation processing is performed. This strategy effectively avoids adhesion issues of distal small branches while ensuring the learnability of vessel length for the model.

**Quality Control.** To further validate the CFD usability of annotations, other than conducting voxel-level segmentation masks, all cases are conducted vascular geometric annotations for CFD analysis, including STL files of cut inlet/outlet sections, vascular centerline data, and mesh grid files labeled with fluid boundary conditions. Besides, CFD applicability of the segmentation masks are evaluated to validate whether the pressure and velocity residuals in the blood flow dynamics analysis achieve convergence. We perform a rigorous quality check and screening, annotations not applicable for CFD are further refined and validated, or removed from the final dataset.

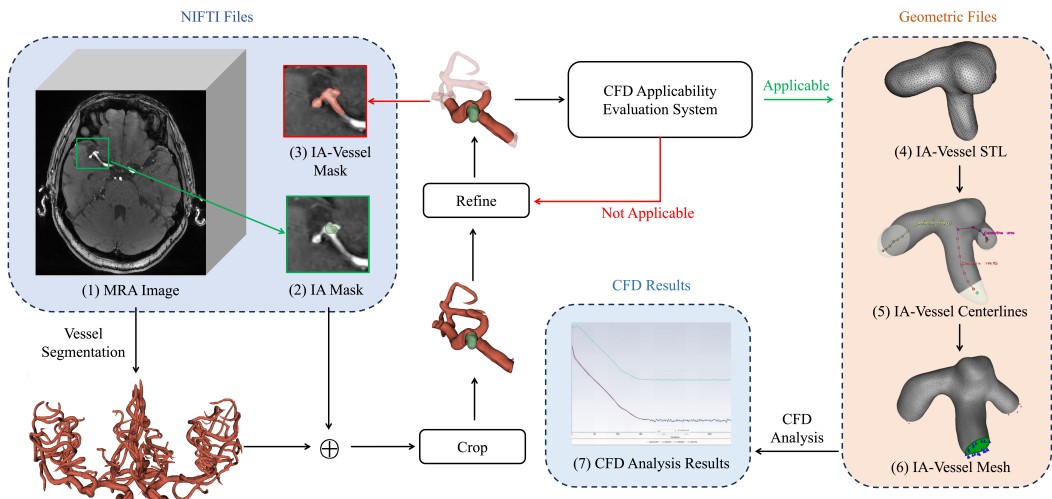

Figure 2: An overview of the IAVS dataset and the annotation workflow. Each case encompasses seven types of standardized data: (1) whole-brain MRA images, (2) IA mask, (3) IA-Vessel mask, (4) STL models with cut inlets/outlets, (5) vascular centerlines, (6) mesh files with boundary annotations, (7) CFD analysis results.

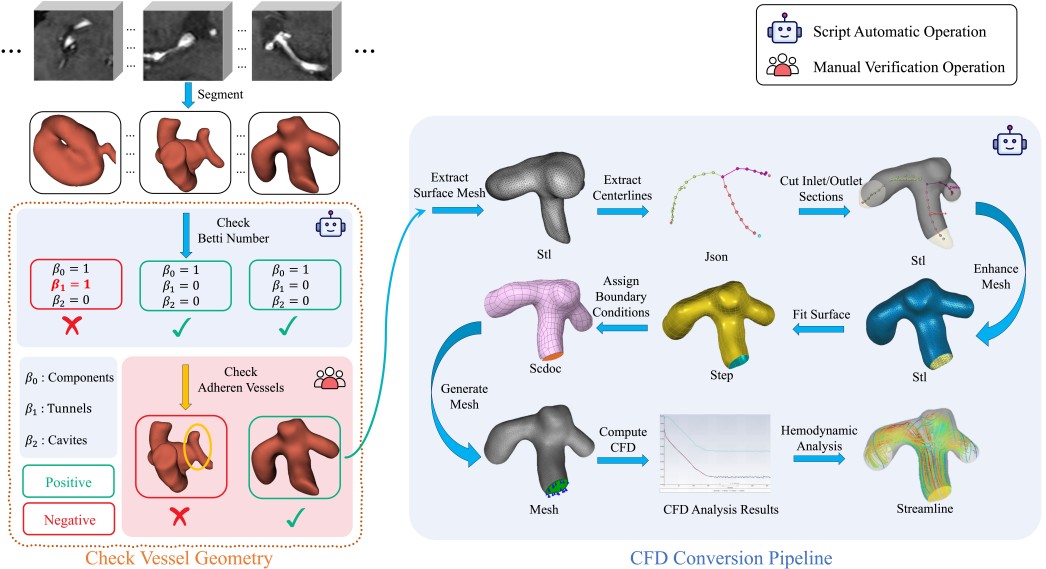

Figure 3: Overview of our conversion pipeline from segmentation masks to CFD models, which realizes the entire chain process from medical imaging to flow field simulation. The pipeline consists of following steps, including vascular topology inspection, morphological preprocessing, geometric model conversion, centerline generation, end face cutting, mesh enhancement, surface fitting, boundary labeling, mesh generation, and CFD computation.

# 4 CFD APPLICABILITY EVALUATION SYSTEM

To achieve automated and standardized conversion from segmentation mask to CFD model, we establish a standardised CFD applicability evaluation system as shown in Figure 3. The pipeline consists

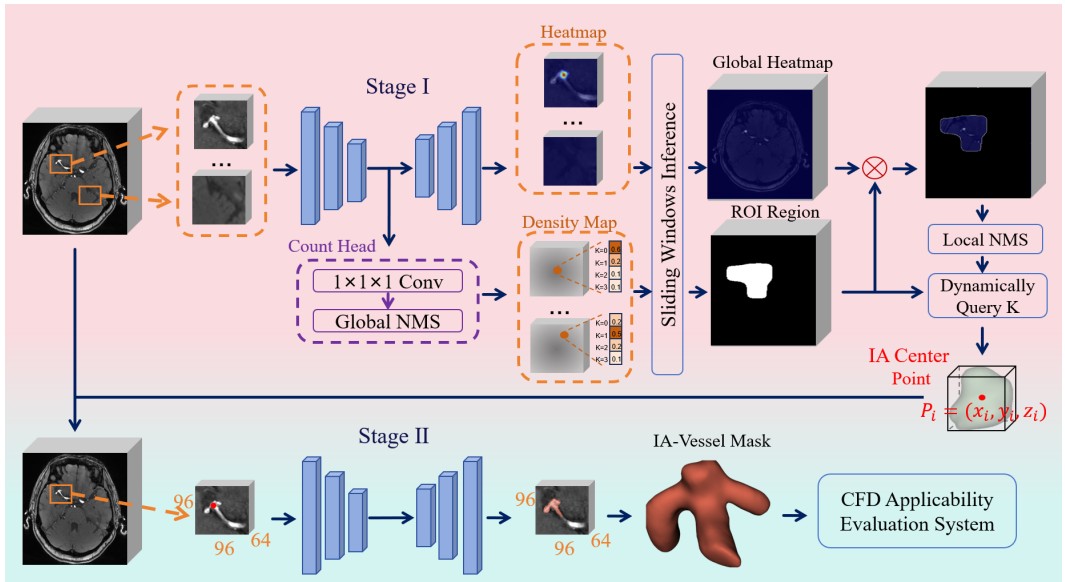

Figure 4: Our proposed two-stage framework for IA-Vessel segmentation. Stage I utilizes a detection network for global localization of aneurysms. After cropping out candidate patches, Stage II utilizes a topological-aware segmentation network for IA-Vessel segmentation to reduce topology errors.

of the following steps, including vascular topology inspection, morphological preprocessing, geometric model conversion, centerline generation, end face cutting, mesh enhancement, surface fitting, boundary labeling, mesh generation, and CFD computation. The detailed procedure for each step is shown in the Appendix A.

Based on the evaluation system, we propose a novel applicability-based evaluation metric entitled CFD applicability score (CFD-AS) to enable more comprehensive evaluation of segmentation results, which is defined as follows:

$$AS_{\text{CFD}} = \frac{\widehat{TP}}{TP + FP + FN} \tag{1}$$

$$\widehat{TP} = \sum_{i=1}^{N} (y = 1) \wedge (\hat{y} = 1) \wedge (AE(\hat{y}) = 1) \tag{2}$$

$$AE_{\hat{y}} = \begin{cases} 1, & \text{if } (VTA_{\hat{y}}) = 1 \wedge (MGA_{\hat{y}}) = 1 \wedge (BFA_{\hat{y}}) = 1 \\ 0, & \text{if } (VTA_{\hat{y}}) = 0 \vee (MGA_{\hat{y}}) = 0 \vee (BFA_{\hat{y}}) = 0 \end{cases} \tag{3}$$

where $\widehat{TP}$ represents true positive cases that can be successfully applicable for CFD analysis. Specifically, $VTA_{\hat{y}} \in \{0, 1\}$, $MGA_{\hat{y}} \in \{0, 1\}$, and $BFA_{\hat{y}} \in \{0, 1\}$ represent the vascular topology availability, mesh generation availability, and blood flow availability of the segmentation mask $\hat{y}$, indicating, respectively, whether there are geometric topological abnormalities in the vessels, whether geometric errors occur during the conversion process that interrupt subsequent operations, and whether the generated mesh file can be successfully used for CFD analysis. The computation of all three indices above can be automated via scripts.

## 5 BENCHMARK DESIGN

To demonstrate the utility of the IAVS dataset, we adopt two benchmark tasks that mirror the clinical workflow from raw MRA to simulation-ready geometry: **Stage I for global localization of**

**aneurysms** and **Stage II for fine-grained IA-Vessel segmentation**. These benchmarks are designed to evaluate methods on clinically meaningful tasks. In Stage I, a global localization step identifies regions of interest containing aneurysms, setting the stage for more precise analysis. Stage II then focuses on fine-grained, topology-aware segmentation of the IA-Vessel within these localized regions. This approach is specifically designed to evaluate segmentation performance in the context of topological consistency and CFD applicability. As illustrated in Figure 4, we develop a simple and effective two-stage framework, which can be used as a out-of-the-box method and strong baseline for the benchmark.

**Stage I: Aneurysm Localization.** To overcome the difficulty of directly segmenting small aneurysms from full MRA volumes, we first use a detection network to pinpoint their locations. We use a counting-guided heatmap formulation to substantially reduce false positives by constraining the predicted count. Specifically, the network is designed to simultaneously predict a heatmap, indicating the probability of an aneurysm center, and a density map, estimating the number of aneurysms. The training loss is shown in Formula 4, which consists of two parts. The first part is the heatmap loss, inspired by the focal loss used for centroid prediction in Zhou et al. (2019). Due to the extreme sparsity of positive voxels (aneurysm centers), each ground truth center point is supervised using a 3D Gaussian heatmap $t_{\text{xyz}}$ with a peak value of 1. To address the severe foreground-background imbalance, a weighting scheme is applied. For positive voxels ($t_{\text{xyz}} \geq 0.9$), the loss is $(1-p_{\text{xyz}})^\alpha \cdot \log(p_{\text{xyz}})$. For all other voxels (negatives), the loss is $(1-t_{\text{xyz}})^\beta \cdot p_{\text{xyz}}^\alpha \cdot \log(1-p_{\text{xyz}})$. Here, $p_{\text{xyz}}$ is the predicted heatmap value, $\alpha$ is a focusing parameter that down-weights easily classified examples, and the $(1 - t_{\text{xyz}})^\beta$ term for negatives places more emphasis on ambiguous regions near the Gaussian boundaries. The total heatmap loss is normalized by the number of positive voxels $N_{\text{pos}}$. The second part is a standard cross-entropy loss for the aneurysm count classification, where the number of aneurysms per case is treated as a classification problem with classes ranging from 0 to 5.

$$\mathcal{L}_{\text{Stage I}} = -\frac{1}{N_{\text{pos}}} \underbrace{\left[ \sum_{x,y,z} \begin{cases} (1-p_{\text{xyz}})^\alpha \log(p_{\text{xyz}}) & \text{if } t_{\text{xyz}} \geq 0.9 \\ (1-t_{\text{xyz}})^\beta p_{\text{xyz}}^\alpha \log(1-p_{\text{xyz}}) & \text{otherwise} \end{cases} \right]}_{\text{Heatmap Loss}} + \underbrace{\mathcal{L}_{\text{CE}}(C_{\text{pred}}, C_{\text{true}})}_{\text{Count Classification Loss}}, \quad (4)$$

During inference, candidate center points are extracted from aggregated heatmaps and density maps. We employ a dynamic selection mechanism where the number of candidates is adaptively determined by the connected components in the density map, effectively reducing false positives. This point-based detection is less sensitive to variations in aneurysm size compared to standard segmentation or bounding-box detection.

**Stage II: IA-Vessel Segmentation.** Using the center points from Stage I, we crop candidate patches to focus the segmentation task. In Stage II, we apply a topology-aware segmentation network built upon the robust nnUNet (Isensee et al., 2021) backbone. To ensure the resulting vessel geometry is suitable for CFD analysis, we incorporate a loss function that preserves vascular connectivity. As shown in Equation 5, the total loss combines a standard segmentation loss (Dice and cross-entropy) with a clDice loss term. The clDice component enhances the model's sensitivity to vascular topology by explicitly supervising on centerline connectivity, which is critical for preventing spurious connections or breaks in the vessel structure.

$$\mathcal{L}_{\text{Stage II}} = \underbrace{-\frac{2\sum_i p_i g_i}{\sum_i p_i + \sum_i g_i} + \left(-\sum_i g_i \log p_i\right)}_{\text{Segmentation Loss}} + \lambda \underbrace{\left(-\frac{2\sum_i \mathcal{T}(p_i)\mathcal{T}(g_i)}{\sum_i \mathcal{T}(p_i) + \sum_i \mathcal{T}(g_i)}\right)}_{\text{clDice Loss}} \quad (5)$$

# 6 EXPERIMENTS

We systematically evaluate the proposed framework compared with existing state-of-the-art methods on the IAVS dataset, including the evaluation of aneurysm detection for Stage I, the evaluation of IA-Vessel segmentation for Stage II, and the comprehensive evaluation of end-to-end segmentation

Table 4: Comparison of different strategies for aneurysm localization in Stage I.

| Method | Set A | | | |
| --- | --- | --- | --- | --- |
| | PR↑ | RE↑ | ACC↑ | F1↑ |
| nnDetection | 0.3737 ± 0.4122 | **0.9250** ± 0.4926 | 0.3627 ± 0.4131 | 0.5324 ± 0.4225 |
| SwinUNETR | 0.3472 ± 0.4718 | 0.6098 ± 0.5004 | 0.2841 ± 0.4644 | 0.4425 ± 0.4642 |
| nnUNet | 0.5778 ± 0.4754 | 0.6341 ± 0.4853 | 0.4333 ± 0.4668 | 0.6047 ± 0.4650 |
| Ours | **0.8286** ± **0.4182** | 0.7073 ± 0.4238 | **0.6170** ± 0.4222 | **0.7632** ± 0.4102 |
| Method | Set B | | | |
| | PR↑ | RE↑ | ACC↑ | F1↑ |
| nnDetection | 0.5440 ± 0.3036 | **0.9292** ± 0.1751 | 0.5224 ± 0.3017 | 0.6863 ± 0.2389 |
| SwinUNETR | 0.5145 ± 0.3956 | 0.7807 ± 0.3792 | 0.4495 ± 0.3884 | 0.6202 ± 0.3650 |
| nnUNet | 0.6942 ± 0.4046 | 0.7368 ± 0.4033 | 0.5563 ± 0.4008 | 0.7149 ± 0.3846 |
| Ours | **0.8785** ± 0.2523 | 0.8246 ± 0.2708 | **0.7402** ± 0.2871 | **0.8507** ± 0.2491 |

with CFD applicability score. We split the test set into Set A for evaluation from multiple public datasets, and Set B for evaluation in clinical scenarios from our private dataset. More experimental and implementation details are shown in the Appendix B.

Table 5: Ablation experiments of topological-aware loss for IA-Vessel segmentation in Stage II.

| Model | Topological-aware Loss | Set A | | | |
| --- | --- | --- | --- | --- | --- |
| | | Dice↑ | HD95↓ | clDice↑ | BIoU↑ |
| Zig-RiR | ✗ | 0.7069 ± 0.1366 | 7.6029 ± 4.2749 | 0.6975 ± 0.1352 | 0.5626 ± 0.1568 |
| nnUNet | ✗ | 0.8533 ± 0.0840 | **3.2187** ± 2.7283 | 0.8555 ± 0.1113 | 0.7527 ± 0.1211 |
| nnUNet | Skeleton Recall Loss | 0.8401 ± 0.0958 | 3.5820 ± 3.4290 | 0.8447 ± 0.1269 | 0.7350 ± 0.1330 |
| nnUNet | clDice Loss | **0.8563** ± 0.0878 | 3.2809 ± 3.0868 | **0.8629** ± 0.1175 | **0.7576** ± 0.1214 |
| Model | Topological-aware Loss | Set B | | | |
| | | Dice↑ | HD95↓ | clDice↑ | BIoU↑ |
| Zig-RiR | ✗ | 0.7536 ± 0.1353 | 6.0273 ± 4.9289 | 0.7425 ± 0.8521 | 0.6216 ± 0.7363 |
| nnUNet | ✗ | 0.8363 ± 0.1307 | 4.2557 ± 5.6929 | 0.8538 ± 0.1502 | 0.7368 ± 0.1652 |
| nnUNet | Skeleton Recall Loss | 0.8296 ± 0.1452 | **4.1835** ± 5.7645 | 0.8516 ± 0.1565 | 0.7303 ± 0.1769 |
| nnUNet | clDice Loss | **0.8368** ± 0.1368 | 4.2134 ± 5.5734 | **0.8616** ± 0.1524 | **0.7388** ± 0.1693 |

## 6.1 EVALUATION OF ANEURYSM DETECTION

To address the challenge of localizing small aneurysms, we conduct a comprehensive evaluation of proposed method with existing approaches. Specifically, we use three different task settings to achieve the localization of aneurysms, including utilizing state-of-the-art detention model nnDetection (Baumgartner et al., 2021), and segmentation models SwinUNETR (Hatamizadeh et al., 2021) and nnUNet (Isensee et al., 2021), where the segmentation results is processed to generate the center point of output targets. As illustrated in Table 4, our proposed method stands out with remarkable performance in multiple metrics. We achieve a PR of 0.8286 and 0.8785, ACC of 0.6170 and 0.7402, and F1-scores of 0.7632 and 0.8507 in Set A and Set B, respectively. Although our method exhibits a slightly lower RE compared to nnDetection, our innovative dynamic candidate point selection mechanism plays a crucial role. This mechanism effectively controls the false positive rate, preventing the generation of an excessive number of false detections. As a result, it alleviates the computational burden and complexity of subsequent processing stages, providing a more efficient and reliable solution for small aneurysm localization. Although heatmap regression itself is a standard technique in keypoint detection, our modification does not aim to introduce a new general localization theory. Instead, to explicitly address the false-positive issue, we design a loss function that combines a heatmap regression loss and a count classification loss. Compared with segmentation-based localization and bounding-box detection, our design is an engineering optimization tailored to this specific task to substantially reduce false positives. Overall, our method significantly outperforms the exist-

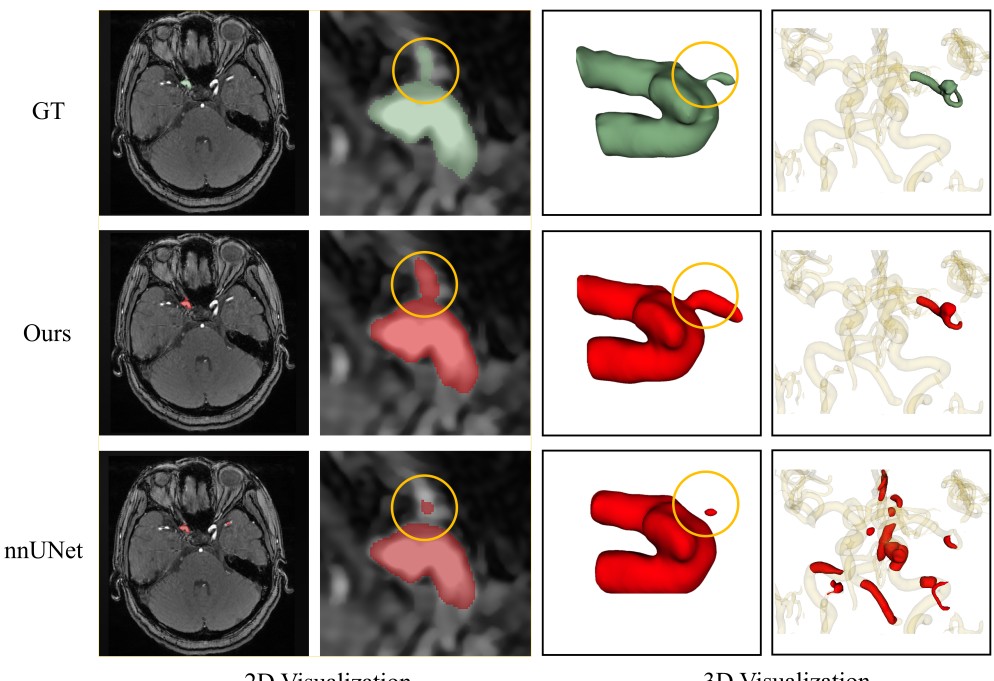

2D Visualization             3D Visualization

Figure 5: Visualization of IA-Vessel segmentation results of different methods.

ing detection and segmentation methods, demonstrating its strong competitiveness and potential for practical applications in medical imaging analysis.

In addition, we conduct new experiments on the publicly available GLIA-Net (Bo et al., 2021) dataset for aneurysm detection. The experimental results are presented in Appendix C. The experiments demonstrate that our method achieves optimal performance across all evaluation metrics.

## 6.2 EVALUATION OF IA-VESSEL SEGMENTATION

To evaluate the effectiveness of proposed topological-aware segmentation framework, we conduct ablations of clDice loss for the segmentation. For the training of Stage II, input patches cropped based on ground truth IA are utilized for localization to avoid error accumulation in Stage I. As observed in Table. 5, the results show that the introduction of clDice loss significantly enhances the vascular topology maintenance ability, which improves the clDice performance from 0.8555 to 0.8629 on Set A and 0.8538 to 0.8616 on Set B.

## 6.3 EVALUATION OF CFD APPLICABILITY

To make a comprehensive evaluation of our framework for CFD applicable IA-Vessel segmentation from MRA images, we integrate the localization results of Stage I with the segmentation procedure of Stage II to enable end-to-end segmentation. In comparison, we conduct direct end-to-end IA-Vessel segmentation using state-of-the-art nnUNet (Isensee et al., 2021) as baseline performance, and ground truth aneurysm localization for patch cropping as an upperbound comparison. As shown in Table 6, end-to-end nnUNet segmentation yields Dice coefficients of 0.1548 on Set A and 0.4557 on Set B. Due to an excessive number of false positives over 120 and fewer than 10 true positives, we conclude that the end-to-end segmentation approach is not suitable for the segmentation task. As shown in Table 7, among comparison of two-stage frameworks, we observe that our method achieve a high applicability score of 57.45% and 54.76%, significantly outperforms other comparing methods by a large margin. As shown in Figure 5, we observe that proposed method can generate mask predictions align more accurately with ground truth masks with less topologic errors and false positive predictions of background vessels.

Table 6: Performance of different methods for end-to-end IA-Vessel segmentation from MRA images.

| Framework | Set A | | | |
|---|---|---|---|---|
| | Dice↑ | HD95↓ | clDice↑ | BIoU↑ |
| nnUNet Baseline | $0.1548 \pm 0.2520$ | $48.8495 \pm 39.9534$ | $0.1552 \pm 0.2468$ | $0.1088 \pm 0.1927$ |
| Stage I nnDetection + Stage II | $0.4285 \pm 0.3753$ | $\mathbf{15.9663} \pm 17.8063$ | $0.4311 \pm 0.3862$ | $0.3477 \pm 0.3236$ |
| Stage I nnUNet + Stage II | $0.4864 \pm 0.4070$ | $50.5446 \pm 90.9340$ | $0.4943 \pm 0.4124$ | $0.4174 \pm 0.3686$ |
| Stage I Ours + Stage II | $\mathbf{0.6324} \pm 0.3630$ | $27.8342 \pm 65.1445$ | $\mathbf{0.6361} \pm 0.3682$ | $\mathbf{0.5482} \pm 0.3356$ |
| Stage I GT + Stage II | $0.8563 \pm 0.0878$ | $3.2809 \pm 3.0868$ | $0.8629 \pm 0.1175$ | $0.7576 \pm 0.1214$ |
| Framework | Set B | | | |
| | Dice↑ | HD95↓ | clDice↑ | BIoU↑ |
| nnUNet Baseline | $0.4557 \pm 0.2898$ | $37.2055 \pm 17.9024$ | $0.4323 \pm 0.2853$ | $0.3395 \pm 0.2496$ |
| Stage I nnDetection + Stage II | $0.6611 \pm 0.2252$ | $19.6505 \pm 17.8079$ | $0.6846 \pm 0.2334$ | $0.5344 \pm 0.2476$ |
| Stage I nnUNet + Stage II | $0.6186 \pm 0.3499$ | $65.7056 \pm 112.9016$ | $0.6477 \pm 0.3556$ | $0.5286 \pm 0.3264$ |
| Stage I Ours + Stage II | $\mathbf{0.7442} \pm 0.2406$ | $\mathbf{15.8149} \pm 36.6947$ | $\mathbf{0.7706} \pm 0.2533$ | $\mathbf{0.6391} \pm 0.2500$ |
| Stage I GT + Stage II | $0.8368 \pm 0.1368$ | $4.2134 \pm 5.5734$ | $0.8616 \pm 0.1524$ | $0.7388 \pm 0.1693$ |

Table 7: Evaluation of CFD Applicability Score of different IA-Vessel segmentation masks.

| Framework | Set A | | | | | Set B | | | | |
|---|---|---|---|---|---|---|---|---|---|---|
| | TP | FP | FN | $\widehat{TP}$ | $AS_{CFD}$ | TP | FP | FN | $\widehat{TP}$ | $AS_{CFD}$ |
| Stage I nnDetection + Stage II | 37 | 62 | 4 | 30 | 29.13% | 105 | 88 | 9 | 74 | 36.63% |
| Stage I nnUNet + Stage II | 26 | 19 | 15 | 23 | 38.33% | 84 | 37 | 30 | 65 | 43.05% |
| Stage I Ours + Stage II | 29 | 6 | 12 | 27 | **57.45%** | 94 | 12 | 20 | 69 | **54.76%** |
| Stage I GT + Stage II | 41 | 0 | 0 | 35 | 85.37% | 114 | 0 | 0 | 88 | 77.19% |
| IA-Vessel GT | 41 | 0 | 0 | 41 | 100.00% | 114 | 0 | 0 | 114 | 100.00% |

# 7 DISCUSSION AND CONCLUSION

In this work, we introduce a systematic solution for CFD-applicable IA-Vessel segmentation. To overcome the limitations of existing datasets, we construct IAVS, a large-scale multi-centre dataset with comprehensive annotations and CFD analysis results, providing a solid foundation for subsequent research. Our proposed two-stage framework for detection and segmentation effectively reduces geometric errors and enhances the CFD usability of segmentation masks, making a breakthrough in improving the accuracy and reliability of segmentation. Additionally, the establishment of a standardized CFD applicability evaluation system, along with the introduction of the CFD applicability score, enables a more comprehensive and standardized evaluation of segmentation results. Experimental results demonstrate that our proposed method achieves a high CFD applicability score of 57.45% and 54.76% on different test sets, which is significantly higher than that of existing state-of-the-art methods, verifying its clinical applicability in CFD analysis, so as to assist in clinical decision-making.

**Limitations.** Firstly, our framework employs independent training procedure of each stage, which may limit further performance improvement of the model. Future work could explore an end-to-end joint training mechanism. By sharing encoder-layer features and jointly optimizing the loss functions, the tasks of localization and segmentation could be synergistically enhanced. Besides, existing loss functions for training segmentation models primarily rely on image-based segmentation metrics, which have a semantic gap with the CFD applicability. Future work could focus on utilizing the applicability-based evaluation for optimization of segmentation networks to enhance the applicability of segmentation results for CFD applications.

Currently, CFD validation needs to be performed independently of the segmentation process. Future research could introduce physics-informed neural networks to build an end-to-end predictive model from segmentation results to hemodynamic parameters, achieving a closed-loop optimization between segmentation and simulation.(Lu et al., 2019; Yao et al., 2024b).

## ETHICS STATEMENT

The authors of this paper have read and adheres to the ICLR Code of Ethics. This research involves the use of sensitive medical data and aims for a direct clinical application; therefore, we have taken several steps to ensure our work is conducted with the highest ethical standards.

**Human Subjects and Data Privacy:** Our study utilizes 3D MRA images from both publicly available and private, in-house clinical datasets. The collection and use of the in-house patient data were conducted in full compliance with institutional and national ethical guidelines. The study protocol, including data collection and anonymization procedures, received formal approval from the relevant hospital's Institutional Review Board (IRB) / Ethics Committee. All data were fully anonymized prior to their use in this research, with all personally identifiable information (PII) removed to protect patient privacy and confidentiality.

**Dataset Curation and Release:** We are committed to scientific transparency and reproducibility. Upon acceptance, the curated IAVS dataset, along with our code and models, will be made publicly available. We will ensure that the released data is thoroughly de-identified to prevent any potential for re-identification of individuals, thereby responsibly contributing a valuable resource to the research community while upholding our duty to protect patient privacy.

**Potential for Societal Impact and Misuse:** The primary goal of this research is to contribute positively to human well-being by improving the accuracy and applicability of intracranial aneurysm segmentation for hemodynamic analysis. This can ultimately aid clinicians in assessing aneurysm rupture risk and making more informed treatment decisions. However, we acknowledge that any automated medical analysis tool carries the risk of misuse if not properly validated and deployed. Our proposed framework is intended to be used as a decision-support tool to assist trained medical professionals (such as radiologists and neurosurgeons) and is not designed to replace clinical expertise or serve as a standalone diagnostic system.

**Bias and Fairness:** Our IAVS dataset is compiled from multiple centers, which helps to mitigate biases associated with a single institution's population or imaging hardware. Nonetheless, the demographic distribution (e.g., race, age, sex) of the patient data may not fully represent the global population. This could potentially lead to performance disparities when the model is applied to underrepresented groups. We acknowledge this as a limitation and advocate for future work to validate and fine-tune our models on more diverse and larger-scale datasets to ensure equitable and robust performance across all patient populations.

## REPRODUCIBILITY STATEMENT

To ensure the reproducibility of our research, we have provided comprehensive details throughout the paper and its appendices. The construction, annotation workflow, and statistical breakdown of our proposed **IAVS** dataset are thoroughly described in Section 3. A detailed description of our proposed two-stage framework, including the network architectures and loss functions for both detection and segmentation stages, is provided in Section 5. All experimental settings, including data preprocessing, training hyperparameters (e.g., optimizer, learning rate, patch sizes), and the specific evaluation metrics used, are detailed in Appendix B. The procedure for our novel CFD applicability evaluation system, which automates the conversion from segmentation masks to CFD models, is outlined step-by-step in Section 4 and Appendix A. As stated in the abstract, we are committed to transparency and will make our source code, the complete IAVS dataset, and the pre-trained models publicly available upon acceptance of this manuscript to facilitate verification and further research in the community.

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

## A   PROCEDURE FOR CFD APPLICABILITY EVALUATION

This study establishes a standardized workflow for transforming medical images into computational fluid dynamics models consists of following steps.

**Vascular Topology Inspection** The vascular topology of the segmentation results of intracranial aneurysms and their associated vessels is first screened to detect geometric defects such as abnormal adhesion, holes, indentations, and protrusions. These voxel-level segmentation errors, although not affecting traditional segmentation metrics such as the Dice coefficient, can significantly impact the integrity of vascular geometry and subsequently cause flow field distortions in CFD analysis. Issues such as discontinuities, holes, and partial adhesions can be identified by computing Betti numbers, which can detect most topological problems, while a minority of other issues still require manual verification.

**Preprocessing of Segmentation Results** Morphological optimization operations are performed using 3D Slicer software, including removal of stretched regions, filling of small holes, and smoothing of details. A median filter with a kernel size of 1 mm is uniformly applied for surface smoothing to eliminate discrete segmentation artifacts while ensuring reproducibility. The largest connected component is extracted after smoothing to exclude isolated noise structures. It should be noted that approximately 1% of samples may experience abnormal adhesion due to excessive smoothing, which requires manual correction using the segmentation tools in 3D Slicer.

**Conversion to Geometric Models** The voxel-represented NIFTI image data is converted into a three-dimensional geometric STL model, providing a geometric basis for subsequent CFD analysis.

**Generation of Vascular Inlet and Outlet Endpoints and Centerlines** Based on the generated STL model, the VMTK toolkit is used to automatically identify the topological endpoints of vascular inlets and outlets, and to generate vascular centerlines accordingly. When automatic detection deviates, interactive corrections are made using 3D Slicer.

**Cutting of Inlet and Outlet Cross-sections** The ptvista library is used to cut vascular cross-sections based on the normal vectors of the centerlines. The cutting plane is uniformly set at the 1/5 end position (when the candidate cutting point radius is less than 0.3 mm, it automatically retracts to a proximal position that meets the radius requirement). This approach retains the complete vascular structure while avoiding morphological distortion caused by excessive cutting. Experiments have shown that approximately 10% of samples fail automatic cutting due to insufficient centerline length, requiring manual intervention using Geomagic Wrap 2021.

**Mesh Enhancement** Geomagic Wrap is used to perform mesh optimization processes: first, the mesh doctor is used to repair non-manifold edges, self-intersections, and highly refractive edges, followed by mesh re-meshing, refinement, optimization, and enhancement operations. All parameters are set to the software's default values to ensure consistency in processing.

**Fitting of Surface Geometry Files** The STL mesh file is reconstructed into a CAD model with precise geometric definitions and topological relationships, i.e., a STEP format file. Based on the STL mesh file, surface patches are constructed, a grid is built, and the surface is fitted to generate the STEP file. The number of surface patches is set to 1000. At this point, less than 1% of the data may detect intersecting grids during grid construction, which can be manually repaired by moving surface patch vertices to eliminate concave polygons. For cases where the surface patches are too large, the patches can be subdivided to resolve the issue.

**Boundary Condition Annotation** The STEP model is imported into ANSYS SpaceClaim for boundary condition definition, including the precise annotation of inlets, outlets, walls, and fluid regions. The end faces are automatically identified using the previously generated endpoints and centerline information, and the final model is saved in SCDOC format.

**Mesh Generation** Fluent Meshing is used to generate unstructured polyhedral meshes, with mesh quality and computational stability ensured through CFL number control and residual monitoring mechanisms.

**CFD Calculation** Blood flow field simulation is performed using the incompressible Newtonian fluid model. The Navier-Stokes equations are solved using the icoFoam solver in Open Field Operation and Manipulation (OpenFOAM (Weller et al., 1998) combined with the PISO algorithm,

calculating the velocity field, pressure field, and wall shear stress distribution under a mass flow rate range of 0.00100.0040 kg/s (only steady-state calculations are performed).

Prior to this, there is no fully automated workflow for converting binary segmentation masks to computational fluid dynamics models. The alternate use of multiple industrial software packages, as well as the cumbersome and repetitive nature of the operational process, significantly increases the labor and time costs associated with the annotation process. Moreover, the subjective variability introduced by manual cutting of vascular inlet and outlet cross-sections directly affects the objectivity of CFD-AS calculations.

## B    EXPERIMENTAL SETTINGS

**Implementation Details.** All of our experiments are implemented in Python with PyTorch, using an NVIDIA A100 GPU. We use the SGD optimizer with an initial learning rate of 0.01, a weight decay of 3e-5 and a momentum of 0.99 to update the network parameters with the maximum epoch number set to 1000. In Stage I, the original images are resampled to a voxel spacing of $0.34 \times 0.34 \times 0.55$ $mm^3$, and then are cropped into patches of size $224 \times 224 \times 48$, with the batch size set to 2. In Stage II, the patch size is $96 \times 96 \times 64$ and the batch size set to 9. During the training stage, random cropping, flipping and rotation are used to enlarge the training set and avoid over-fitting. In the inference stage, the final segmentation results are obtained using a sliding-window strategy. For other comparing methods, we follow the official implementations as in (Hatamizadeh et al., 2021; Baumgartner et al., 2021).

**Evaluation Metrics.** For aneurysm detection, four metrics include precision (PR), recall (RE), average precision (AP), and the F1 score are used for evaluation. These metrics are defined as follows:

$$PR = \frac{TP}{TP + FP}$$

$$RE = \frac{TP}{TP + FN}$$

$$Acc = \frac{TP}{TP + FP + FN}$$

$$F1 = 2 \times \frac{PR \times RE}{PR + RE}$$

where TP, FP, TN, FN represent true positive (correct detection of an aneurysm), false positive (incorrect detection of an aneurysm in a healthy case), and false negative (missed detection of an aneurysm), respectively.

For IA-Vessel segmentation, four metrics including the Dice similarity coefficient (Dice), 95% Hausdorff Distance (HD95), the centerline Dice (clDice), and the boundary IoU (BIoU) are used for evaluation. These metrics are defined as follows:

$$Dice = \frac{2 \times |A \cap B|}{|A| + |B|}$$

$$HD95 = \inf \{d \geq 0 \mid S_A \subseteq \mathcal{N}_d(S_B) \text{ and } S_B \subseteq \mathcal{N}_d(S_A)\}$$

$$clDice = \frac{2 \times |C_A \cap C_B|}{|C_A| + |C_B|}$$

$$BIoU = \frac{|\partial A \cap \partial B|}{|\partial A \cup \partial B|}$$

where $A$ and $B$ represent the predicted segmentation mask and ground truth. $\mathcal{N}_d(\cdot)$ denotes the $d$-neighborhood around a set, and $S_{\text{pred}}/S_{\text{gt}}$ are the predicted/ground truth boundaries. $C_A/C_B$ and

$\partial A / \partial B$ represent the centerline and the boundary pixels of predicted segmentation mask and ground truth, respectively.

## C ADDITIONAL EXPERIMENTS

We further demonstrat that the heatmap cascading strategy of detection can effectively enhance segmentation performance. When the encoder pretrained in our framework are used to initialize the encoder of nnUNet for aneurysm segmentation, the Dice coefficient of the segmentation model increases from 0.4841 to 0.5676 as shown in Figure 8, confirming the position constraint effect of localization on aneurysms. Ablation experiments show that fixing the encoder of keypoint detection for feature transfer outperforms the direct cascading of heatmaps (Dice: 0.5676 vs. 0.5148), indicating that semantic consistency in the feature space is crucial for segmentation accuracy.

Table 8: Impact of keypoint detection on segmentation performance

| Model | Set A | | | | Set B | | | |
|---|---|---|---|---|---|---|---|---|
| | Dice↑ | HD95↓ | PR↑ | RE↑ | Dice↑ | HD95↓ | PR↑ | RE↑ |
| SwinUNETR | 0.2911 | **28.82** | 0.6298 | 0.4358 | 0.4780 | 41.04 | 0.6374 | 0.4611 |
| nnUNet | 0.3915 | 43.09 | 0.6029 | 0.5056 | 0.5490 | 58.51 | 0.5904 | 0.5734 |
| Heatmap Cascade | 0.4776 | 48.01 | 0.6079 | 0.4736 | 0.5772 | 50.69 | 0.6455 | 0.5830 |
| Fixed nnUNet encoder | 0.3874 | 33.68 | 0.6587 | **0.5763** | 0.5765 | 40.32 | 0.6363 | **0.6008** |
| Fixed keypoint encoder | **0.5041** | 33.01 | **0.6901** | 0.5586 | **0.5999** | **32.56** | **0.6902** | 0.5945 |

To evaluate the generalization ability of our framework, we conduct extensive experiments on the publicly available GLIA-Net (Bo et al., 2021) dataset for aneurysm detection. The internal dataset includes 1338 3D CTA images/1489 IAs from 6 institutions. The external dataset includes 138 3D CTA images/101 IAs from 2 institutions. After locally retraining nnUNet and our detector, and using the publicly released segmentation weights from GLIA-Net, the experimental results summarized in Table 9 show that our method outperforms existing approaches across all evaluation metrics on the external test set A, external test set B, and internal test set as defined by the GITA-Net official split.

## D ADDITIONAL DISCUSSION

### D.1 RATIONALE FOR LOCALIZED HEMODYNAMIC ANALYSIS

Our decision to focus the CFD simulations on the aneurysm and its adjacent parent vessels, rather than the entire segmented vascular network, represents a principled balance between clinical relevance and computational feasibility. This approach aligns with established standards in 3D aneurysm research, where focusing on local geometry is the mainstream methodology. This is because aneurysm rupture risk assessment primarily depends on local hemodynamic parameters such as pressure, velocity, and Wall Shear Stress (WSS) rather than the global flow dynamics of the complete network. Numerous studies have validated that such local vessel models are sufficient for revealing the key hemodynamic mechanisms essential for pathological analysis. Furthermore, simulating the entire cerebrovascular network, which contains thousands of arterial branches, imposes a prohibitive computational burden that typically requires weeks of supercomputing resources. Such demands render global simulations infeasible for large-scale datasets or clinical translation. By adhering to this focused approach, we ensure that our evaluation pipeline remains computationally efficient, reducing simulation times to hours while maintaining the hemodynamic fidelity necessary for robust risk assessment and the development of physics-informed metrics.

### D.2 NUMERICAL IMPLEMENTATION AND SENSITIVITY ANALYSIS

To ensure the reliability and reproducibility of our CFD results, we adhered to strict engineering assumptions and conducted comprehensive sensitivity analyses regarding mesh generation and solver convergence. We utilized the PISO algorithm within the OpenFOAM framework for pressure-velocity coupling, enforcing a strict Courant-Friedrichs-Lewy (CFL) number below 1 to ensure

Table 9: Comparison of the three test sets in the GLIA-Net dataset.

| Metric | External test A | | | External test B | | | Internal test | | |
|---|---|---|---|---|---|---|---|---|---|
| | nnUNet | GLIA-Net | Ours | nnUNet | GLIA-Net | Ours | nnUNet | GLIA-Net | Ours |
| PR↑ | 0.0631 | 0.3008 | **0.3866** | 0.0354 | 0.3544 | **0.3981** | 0.0681 | 0.4076 | **0.4925** |
| RE↑ | 0.5200 | 0.7400 | **0.9200** | 0.2941 | 0.5490 | **0.8039** | 0.4444 | 0.7698 | **0.7857** |
| ACC↑ | 0.0596 | 0.2721 | **0.3740** | 0.0326 | 0.2745 | **0.3628** | 0.0628 | 0.3633 | **0.4342** |
| F1↑ | 0.1126 | 0.4277 | **0.5444** | 0.0632 | 0.4308 | **0.5325** | 0.1181 | 0.5330 | **0.6055** |

numerical computation stability and convergence. Convergence was defined by residuals for velocity components $(u, v, w)$ stabilizing at $10^{-6}$ and pressure $(p)$ residuals below $10^{-6}$. Furthermore, we performed a grid independence study to validate our meshing strategy, testing schemes with minimum element sizes ranging from $0.30$ mm to $0.10$ mm. Our analysis demonstrated a clear convergence trend; specifically, refining the mesh from $0.15$ mm to $0.10$ mm resulted in a negligible relative difference in Wall Average Shear Stress of approximately $0.25\%$. Consequently, we adopted the $0.15$ mm scheme as the standard for our dataset, confirming that our hemodynamic outputs are grid-independent and not artifacts of discretization errors.

### D.3 Assumptions and Limitations of Boundary Conditions

While our simulation pipeline is rigorous, we acknowledge certain limitations inherent to the retrospective nature of the dataset, particularly regarding patient-specific boundary conditions. Ideally, inlet flow rates should be derived from individual phase-contrast MRI or Doppler ultrasound measurements. However, since our dataset is constructed from standard non-invasive MRA scans, patient-specific inlet velocity data was not available. To address this, we adopted a standardized average inlet flow rate as the inlet boundary condition, combined with zero-pressure outlet and no-slip wall conditions. Although this simplification prevents the precise replication of an individual patient's absolute flow dynamics, it is a necessary and widely accepted approximation in large-scale computational studies where non-invasive data collection is prioritized. This approach remains robust for evaluating relative hemodynamic patterns and training deep learning models to learn generalizable physics-based features.

## E Use of Large Language Models

In accordance with the conference guidelines, we disclose the use of Large Language Models (LLMs) in the preparation of this manuscript.

The role of LLMs is strictly limited to that of an assistive tool for language editing and proofreading. Specifically, LLMs were utilized to improve grammar, correct spelling, refine sentence structure for better clarity, and suggest alternative word choices to enhance the overall readability of the text.

The LLMs were not used for research ideation, generation of the core scientific content, methodology design, data analysis, or drawing conclusions. All conceptual and scientific contributions presented in this paper are exclusively the work of the human authors. The authors have carefully reviewed and edited all text and take full responsibility for the scientific accuracy and integrity of the final manuscript.

