# OpenReview forum: "Rethinking Intracranial Aneurysm Vessel Segmentation: A Perspective from Computational Fluid Dynamics Applications"
_ICLR.cc/2026/Conference — Submitted to ICLR 2026_

### Official Review · Reviewer_oRdZ · 2025-10-27

**Soundness:** 3
**Presentation:** 3
**Contribution:** 2
**Rating:** 4
**Confidence:** 4

**Summary:**

This paper introduces IAVS, a new multi-center dataset consisting of 641 3D MRA images with annotations for aneurysms and their parent vessels, along with computational fluid dynamics (CFD) analysis results. It further proposes a two-stage framework:
1.	A detection network using heatmaps and dynamic candidate selection for global aneurysm localization.
2.	A topology-aware segmentation network with clDice supervision for fine-grained IA-vessel segmentation.
To bridge the gap between image segmentation and hemodynamic analysis, the authors develop an automated CFD applicability evaluation system and propose a new metric—CFD-Applicability Score (CFD-AS)—to assess whether segmentation results can be successfully used for CFD simulations.

**Strengths:**

•  Clear motivation: The paper identifies an under-addressed but practically relevant gap between conventional segmentation evaluation and CFD usability in intracranial aneurysm analysis.
•  Comprehensive dataset: IAVS is larger and more structured than prior public datasets (e.g., ADAM, Royal), with annotations extending beyond masks to centerlines, STL models, and CFD convergence results.

**Weaknesses:**

1）Limited methodological novelty (major)
•  The proposed pipeline mainly combines existing components: focal loss for detection, nnUNet backbone for segmentation, and clDice for topology preservation.
•  The CFD applicability metric essentially checks meshability and simulation convergence, which is practical but not conceptually innovative.

2）Evaluation focused on internal dataset
•  Most experiments rely on the newly built IAVS dataset.
•  Although some external tests (e.g., GLIA-Net) are mentioned, there is no convincing demonstration of generalization across domains or acquisition protocols, which is critical for medical applications.

3）Benchmarking and comparison are insufficient
•  While several baselines are included (nnUNet, SwinUNETR, nnDetection), stronger recent methods (transformers, foundation models, SAM-like methods) are missing.

4）CFD-AS metric validity is weakly justified
•  The proposed CFD-Applicability Score is a binary feasibility check (topology, mesh generation, flow convergence). This is useful, but it lacks theoretical grounding or validation regarding hemodynamic accuracy, which should be the ultimate target.

5）Lack of deeper learning or modeling insights
•  The paper does not propose new learning algorithms, loss formulations, or training paradigms tied to CFD properties.
•  It could have explored physics-informed learning or differentiable CFD integration but remains purely post-hoc in its evaluation.

**Questions:**

1. It will be better to strengthen baseline comparisons by including more recent and competitive segmentation and detection models, such as transformer-based approaches or foundation models.
2. It will be necessary to include external validation on independent datasets to better evaluate the generalization ability of the proposed framework beyond the IAVS dataset. Demonstrating performance on different imaging sources or clinical settings would significantly strengthen the paper.
3. It will be better to correlate the CFD-Applicability Score (CFD-AS) with actual CFD parameter deviations (e.g., wall shear stress, flow velocity) rather than only reporting binary feasibility. This would provide stronger evidence that the proposed metric reflects real hemodynamic reliability.
4. It will be better to incorporate CFD considerations into the training process, for example by introducing physics-informed constraints or differentiable surrogate models. Such an approach could enhance both the novelty and the clinical relevance of the work.

---

> ### Author Response · Authors · 2025-11-23
> **Response to Reviewer oRdZ Q1-Q3**
>
> We thank the reviewer for the constructive criticism and detailed suggestions. We appreciate the recognition of our dataset’s scale and the practical utility of our pipeline. Below, we address the specific concerns regarding methodological novelty, external validation, benchmarking, and the depth of CFD integration.
>
> **1. Methodological Novelty**
>
> **Re:**
> We acknowledge that the methodological novelty of the segmentation network architecture itself is modest, as the pipeline is designed to serve as a robust, out-of-the-box baseline. However, we respectfully emphasize that the primary contribution of this work is twofold:
>
> As the primary focus of our work is a new large-scale multi-center dataset  with full geometric representations (STL surfaces, vessel centerlines, watertight meshes, validated CFD results) for 641 volumes, supporting clinically relevant hemodynamic analysis and a standardized CFD applicability system quantifying real-world utility of segmentation outputs, exposing failure modes beyond voxel-wise accuracy. This combination provides, for the first time, a unified platform where segmentation methods can be evaluated both on image metrics and downstream CFD applicability.
>
> We believe providing a unified platform where segmentation methods can be evaluated on both image metrics and downstream CFD applicability fills a critical gap in the community.
>
> **2. External Validation**
>
> **Re:**
> We thank the reviewer for this comment. We would like to clarify the "internal" nature of our validation. The IAVS dataset is inherently **multi-center**, collected from varying scanners and protocols. Our validation experiments were conducted across these distinct subsets to ensure robustness against domain shifts within the dataset.
>
> Regarding external validation on independent public datasets: Currently, there are **no other publicly available datasets** that possess the comparable scope, annotation quality, and specifically, the *validated CFD ground truth* required to evaluate our proposed CFD metrics. The uniqueness of IAVS is precisely what makes external validation on legacy datasets infeasible (as they lack the necessary geometric fidelity for physics simulation). We plan to explore cross-dataset generalization in future studies as more high-quality data becomes available.
>
> **3. Benchmarking and Baselines**: Stronger recent methods (transformers, foundation models, SAM-like methods) are missing.
>
> **Re:**
> We appreciate the suggestion to strengthen the baselines.
> * **Transformers:** Our comparison already includes **SwinUNETR**, a widely adopted transformer-based architecture for 3D medical segmentation.
> * **Foundation Models (SAM):** We clarify that the Segment Anything Model (SAM) is designed for *interactive, prompt-based* segmentation. Since our benchmark focuses on **fully automatic** aneurysm detection and IA-Vessel segmentation, interactive models fall outside the current scope.
> * **Other State-of-the-Art Method:**   To address your concern, we have implemented and evaluated Zig-RiR [1], a very recent RWKV-based method. As shown in the table below, nnUNet significantly outperforms this more recent architecture.
> It is also worth noting that in the field of medical image segmentation, **more recent architectures do not necessarily equate to stronger performance**. nnU-Net continues to be the state-of-the-art method and the champion solution in numerous recent international challenges, demonstrating its exceptional robustness and efficacy [2].
>
> | Method | Set A - Dice | Set A - HD95 | Set A - clDice | Set A - BIoU | Set B - Dice | Set B - HD95 | Set B - clDice | Set B - BIoU |
> | :--- | :--- | :--- | :--- | :--- | :--- | :--- | :--- | :--- |
> | nnUNet | **0.8563** | **3.2809** | **0.8629** | **0.7576** | **0.8368** | **4.2134** | **0.8616** | **0.7388** |
> | Zig-RiR | 0.7069 | 7.6029 | 0.6975 | 0.5626 | 0.7536 | 6.0273 | 0.7425 | 0.6216 |
>
> **Reference**
>
>   [1] Chen, Tianxiang, et al. "Zig-rir: Zigzag rwkv-in-rwkv for efficient medical image segmentation." IEEE Transactions on Medical Imaging, 2025.
>
>   [2] Isensee, Fabian, et al. "nnu-net revisited: A call for rigorous validation in 3d medical image segmentation." International Conference on Medical Image Computing and Computer-Assisted Intervention. Cham: Springer Nature Switzerland, 2024.

---

> > ### Author Response · Authors · 2025-11-23
> > **Response to Reviewer oRdZ Q4-Q5**
> >
> > **4. Validity of CFD-AS Metric**: The CFD-Applicability Score is a binary feasibility check. It lacks validation regarding hemodynamic accuracy (e.g., WSS deviations).
> >
> > **Re:**
> > We thank the reviewer for this excellent and insightful suggestion.  We agree that a quantitative metric correlating segmentation quality directly to deviations in hemodynamic parameters (like WSS) would be a powerful and ultimate goal.
> > It is correct that our current CFD-AS is designed as a foundational feasibility and stability check. Its primary purpose is to answer the critical first question: "Is this segmentation an acceptable input for any stable simulation?"
> >
> > In our view, this binary check is a necessary prerequisite.  A segmentation that fails the CFD-AS (e.g., incorrect topology, mesh failure) cannot produce any meaningful hemodynamic data.  Conversely, a "pass" in our CFD-AS explicitly signifies that the model successfully passed rigorous numerical stability validation, including CFL number control and puvw residual convergence as shown in the following Table.  Therefore, a "pass" is **not just a binary check**. It is a guarantee of numerical stability, which is the essential foundation for hemodynamic accuracy.
> >
> > | Variable | Physical Quantity | Final Stable Residual Level |
> > | :--- | :--- | :--- |
> > | $u, v, w$ | Velocity Components | $10^{-6}$ |
> > | $p$ | Pressure | $< 10^{-6}$ |
> >
> > We believe establishing this robust "pass/fail" stability check is a critical first step.  We will, however, add this point to our discussion, highlighting that a future quantitative correlation, as the reviewer suggests, is a valuable and sophisticated direction for subsequent research.
> >
> >
> >
> >
> > **5. Physics-Informed Learning**: It will be better to incorporate CFD considerations into the training process, for example by introducing physics-informed constraints or differentiable surrogate models.
> >
> > **Re:**
> > We thank the reviewer for this forward-looking suggestion. The concept of integrating physics-informed constraints or differentiable CFD surrogates directly into the segmentation training process is indeed a primary goal for this field.
> >
> > However, moving from our current "post-hoc" evaluation pipeline to a fully end-to-end model presents formidable and non-trivial challenges. First, differentiable CFD solvers are still an emerging technology, often struggling with the stability and computational cost required to handle complex, patient-specific 3D geometries.
> >
> > Second, and more fundamentally, such physics-informed (e.g., PINN) or end-to-end approaches require sparse, real-world ground-truth data (e.g., measured velocity or pressure) within the flow domain to serve as a loss function. This type of data is not available from standard clinical MRA/CTA scans.
> >
> > Our current work establishes a robust and validated "segmentation-to-simulation" pipeline, which we consider a **crucial foundational first step**. We are actively working to address this next challenge. As part of our ongoing research, we are in the process of collecting in-vitro Particle Image Velocimetry (PIV) data from phantom models. This PIV data will provide the sparse, intra-domain velocity measurements necessary to train and validate a true physics-informed model. This remains a significant but high-impact future direction for our work.
> >
> > We hope these responses clarify the positioning of our work as a foundational benchmark that enables valid CFD simulation from medical images.

---

### Official Review · Reviewer_j58o · 2025-10-31

**Soundness:** 3
**Presentation:** 3
**Contribution:** 3
**Rating:** 4
**Confidence:** 4

**Summary:**

The paper introduces the IAVS dataset and a two-stage IA-Vessel segmentation framework, as well as a CFD applicability evaluation system and the CFD-AS metric, which aim to bridge the gap between image segmentation and subsequent CFD simulation in terms of practical usability. In its current version, the paper shows innovation and application-oriented design in dataset construction, evaluation design, and experimental results, but there remain several aspects needing improvement, particularly in methodological details, completeness of experimental settings, reproducibility, and systematic analysis of clinical applicability.

**Strengths:**

1.	The proposed IAVS dataset covers multimodal annotations required for the complete workflow from imaging to CFD (3D MRA images, IA/IA-Vessel masks, STL models, centerlines, meshes, CFD analysis results, etc.), which helps promote standardization and reproducibility of end-to-end research.
2.	By establishing a CFD usability evaluation system and the CFD-AS metric, the study quantifies the “CFD usability” of segmentation results, bridging the gap between common segmentation metrics (e.g., Dice) and real-world applicability.
3.	Stage I’s global localization aids in detecting small aneurysms and surrounding vessels, while Stage II’s topology-aware segmentation (integrating clDice) seeks to preserve vascular connectivity and reduce topological errors that negatively affect meshing and subsequent CFD analysis — a direct optimization for vascular anatomical complexity.
4.	The integration of the segmentation framework with CFD usability evaluation demonstrates that the proposed end-to-end method outperforms conventional segmentation baselines in terms of CFD applicability.

**Weaknesses:**

1.	Lacks systematic analysis of the coupling between Stage I and Stage II and of the feasibility of end-to-end training; there is no quantitative assessment of error propagation or robustness between the two stages.
2.	Although topological constraint losses (e.g., clDice) show improvements in vascular topology, the paper lacks detailed robustness analysis and quantitative results under different topological abnormalities (adhesion, distal branch misalignment, branch disconnection, etc.). Overemphasis on topological connectivity might sacrifice local geometric accuracy; systematic evaluation and recommended weight ranges should be provided.
3.	The engineering assumptions involved in converting segmentation masks to CFD models (meshing, boundary conditions, material parameters, etc.) should be listed and subjected to sensitivity analysis; otherwise, the stability and generalizability of the results are difficult to assess.
4.	Details on the division and statistical analysis of Set A and Set B are insufficient, lacking disclosure of sample sizes, confidence intervals, and significance tests, which limits the credibility of the conclusions.
5.	The paper does not sufficiently demonstrate empirical evidence linking CFD metrics to clinical outcomes (e.g., rupture risk, treatment decisions), which weakens its direct persuasiveness for clinical decision making

**Questions:**

1.	How does the global localization error in Stage I affect the segmentation quality and final CFD usability in Stage II? Could end-to-end joint training or co-optimization alleviate error propagation?
2.	While introducing topological constraints (e.g., clDice) benefits vascular connectivity, can it cause local geometric distortions in certain anatomical variations? Is there a lack of systematic sensitivity analysis for the weighting settings?
3.	Is the detailed conversion process from segmentation to CFD model (mesh generation, boundary conditions, material properties, convergence criteria, etc.) fully disclosed? Are patient-specific parameters reproducible?
4.	Have differences in imaging parameters, resolution, and noise among multi-center data been adequately controlled? Is there sufficient evidence of multi-center generalization capability?
5.	Does the study include direct validation of the CFD results’ clinical relevance (e.g., correlation with rupture risk or treatment decisions)? Is there an evaluation of time cost and workflow stability?
6.	Are the sample sizes of Set A and Set B sufficient? Are statistical indicators such as confidence intervals, significance levels, and effect sizes fully reported?

---

> ### Author Response · Authors · 2025-11-24
> **Response to Reviewer j58o Q1-Q2**
>
> We sincerely thank the reviewer for the detailed critique and insightful questions. We have conducted additional experiments and analyses to address each of your concerns.
>
> **1. Coupling Between Stages and Error Propagation**
>
> **Re:**
> We appreciate this important question. We acknowledge that we have not yet systematically analyzed the coupling or error propagation between the two stages in the current manuscript. As discussed, we plan to implement a fully joint training framework in future work to rigorously evaluate feasibility and robustness. In the revised manuscript, we will add a discussion acknowledging this limitation and outlining it as a key direction for future research.
>
> **2. Robustness of Topological Constraints (clDice)**: Overemphasis on topological connectivity might sacrifice local geometric accuracy. Is there a sensitivity analysis for the weighting settings?
>
> **Re:**
> We thank the reviewer for the observations regarding the potential trade-off between topological connectivity and local geometric fidelity. We have conducted a systematic sensitivity analysis to address this concern and identify the optimal weighting strategy.
>
> We defined the total loss as $\alpha$ segmentation loss + $(1-\alpha)$ topological-aware clDice. We varied $\alpha$ from 1 (no topological constraint) to $1/6$ (strong topological constraint).
>
> As shown in the table below, we identified that α=1/2 yields the most robust performance.
> We confirm the reviewer's hypothesis that excessive emphasis on topological constraints can compromise local geometric accuracy, as observed in our experiments where an overly dominant topological weight led to a decrease in Boundary IoU (BIoU), indicating surface irregularities caused by aggressive connectivity enforcement. However, our analysis demonstrates that a balanced configuration (α=1/2) effectively avoids this trade-off. Instead of inducing distortion, it actually enhances local geometric precision.
>
> **Table: Sensitivity Analysis of Loss Weighting $\alpha$**
>
> | Method | Weight | Set A - Dice | Set A - HD95 | Set A - clDice | Set A - BIoU | Set B - Dice | Set B - HD95 | Set B - clDice | Set B - BIoU |
> | :--- | :--- | :--- | :--- | :--- | :--- | :--- | :--- | :--- | :--- |
> | nnUNet | $\alpha = 1$ | 0.8533 | 3.2187 | 0.8555 | 0.7527 | 0.8363 | 4.2257 | 0.8538 | 0.7368 |
> | **nnUNet** | **$\alpha = 1/2$** | **0.8563** | 3.2809 | 0.8629 | **0.7576** | 0.8368 | 4.2134 | 0.8616 | **0.7388** |
> | nnUNet | $\alpha = 1/3$ | 0.8523 | **3.0824** | **0.8659** | 0.7523 | **0.8370** | **4.0859** | 0.8560 | **0.7388** |
> | nnUNet | $\alpha = 1/6$ | 0.8485 | 3.2180 | 0.8610 | 0.7483 | 0.8345 | 4.3551 | **0.8625** | 0.7358 |

---

> ### Author Response · Authors · 2025-11-24
> **Response to Reviewer j58o Q3-Q4**
>
> **3. Engineering Assumptions and CFD Sensitivity**
>
> **Re:**
> We thank the reviewer for their valuable feedback. We have indeed validated the key engineering assumptions and will supplement and clarify these details in the manuscript as suggested.
>
> We conducted a detailed sensitivity analysis (grid independence validation) on meshing, as summarized in Response Table 1. We tested four schemes with varying element counts. To assess convergence, we compared the Wall Average Shear Stress (WSS) of each scheme against the finest converged scheme (Scheme 4, 0.10 mm).
>
> The analysis showed a clear convergence trend. As the mesh was refined from 0.3 mm (4.26% difference) to 0.2 mm (1.61% difference), the error decreased. Crucially, when the minimum element size reached 0.15 mm (Scheme 3), the relative difference compared to the 0.10 mm scheme dropped to approximately 0.25%. This negligible change demonstrates that the solution had reached grid independence.
>
> Our convergence criteria are fully disclosed in Response Table 2. A simulation was considered converged when the residuals for the velocity components (u, v, w) stabilized at the $10^{-6}$ level, and the pressure (p) residuals stabilized below $10^{-6}$. Furthermore, we defined the boundary conditions as a mass flow inlet, a zero-pressure outlet, and no-slip walls. The PISO algorithm in OpenFOAM was used for pressure-velocity coupling, ensuring the CFL number remained strictly below 1.
>
> The reviewer raised an important point regarding the reproducibility of patient-specific parameters, particularly the inlet flow rate. We must acknowledge this as a limitation. Our data was collected from non-invasive MRA, which cannot simultaneously measure patient-specific inlet velocities non-invasively. Therefore, we could not achieve full patient-specific parameter reproducibility in the current dataset. As an alternative, we adopted an average inlet flow rate as the inlet condition. This is a necessary and reasonable simplification in large-scale CFD studies lacking individual measured data.
>
> We fully agree with this limitation and will state it clearly in the discussion section of the paper. In future work, we plan to collect patient-specific flow data (e.g., using techniques like DSA) to achieve higher-fidelity simulations.
>
> **Response Table 1**
> | Scheme | Min Element Size (mm) | Total Cells | Wall Avg. WSS (Pa) | Relative Difference |
> | :--- | :--- | :--- | :--- | :--- |
> | 1 | 0.3 | 141,349 | 4.4986 | 4.26% |
> | 2 | 0.2 | 207,555 | 4.3844 | 1.61% |
> | 3 | 0.15 | 262,184 | 4.3256 | **0.25%** |
> | 4 | 0.10 | 342,566 | 4.3148 | Baseline |
>
> **Response Table 2**
> | Variable | Physical Quantity | Final Stable Residual Level |
> | :--- | :--- | :--- |
> | $u, v, w$ | Velocity Components | $< 10^{-6}$ |
> | $p$ | Pressure | $< 10^{-6}$ |
>
>
> **4. Clinical Relevance and Workflow Efficiency**
>
> **Re:**
>   We thank the reviewer for their valuable feedback. We have indeed evaluated our workflow's clinical relevance via hemodynamic metrics as well as its time cost and stability.
>
> The primary objective of this study is to validate the effectiveness of our segmentation method as a high-fidelity input for downstream CFD analysis. Therefore, our validation focuses on demonstrating how segmentation accuracy directly enhances the reliability of key downstream hemodynamic metrics. While we did not directly correlate our CFD results with patient clinical endpoints (such as rupture status), our CFD dataset includes key indicators widely recognized in the literature as highly correlated with clinical risk (e.g., Wall Shear Stress (WSS), pressure fields, and velocity fields). Our argument is that precise segmentation is a prerequisite for obtaining reliable CFD metrics. Thus, by confirming our method produces more accurate downstream indicators, we confirm its clinical relevance.
>
> Regarding efficiency and robustness, we also evaluated the workflow. On an HPC supercomputing platform, using 24-core CPU parallel processing, the computation time to complete one steady-state CFD simulation is approximately 2 hours. We ensure stability through two key mechanisms: first, strictly controlling the CFL number to remain less than 1 to guarantee numerical stability; and second, continuously monitoring the residual convergence curves (as shown in Response Table 2) to ensure all simulations meet preset standards. This dual mechanism ensures the stability and reliability of our process.

---

> > ### Author Response · Authors · 2025-11-24
> > **Response to Reviewer j58o Q5-Q6**
> >
> > **5.  Details on the division and statistical analysis of Set A and Set B are insufficient.**
> >
> > **Re:**
> > We thank the reviewer for highlighting the need for rigorous statistical reporting to ensure the credibility of our conclusions. We have addressed this by clarifying the dataset division and supplementing the results with comprehensive statistical tests.
> >
> > **Dataset Division and Sample Sizes:**
> > As detailed in Revised **Table 2** of the manuscript shown below, the division of our testing sets is designed to evaluate generalization across different data sources:
> >
> > **Statistics of the proposed IAVS dataset including data source, number and diameter of IAs.**
> >
> > | Dataset | Images (Total) | Images (Public) | Images (Private) | IAs/Case (0) | IAs/Case (1) | IAs/Case (2) | IAs/Case (≥3) | Total IAs | Dia (<3mm) | Dia (3-7mm) | Dia (>7mm) |
> > | :--- | :---: | :---: | :---: | :---: | :---: | :---: | :---: | :---: | :---: | :---: | :---: |
> > | **Set A** | 76 | 76 | 0 | 42 | 29 | 3 | 2 | 41 | 16 | 17 | 8 |
> > | **Set B** | 98 | 0 | 98 | 0 | 85 | 10 | 3 | 114 | 10 | 93 | 11 |
> >
> > **Enhanced Statistical Analysis:**
> > We acknowledge that the initial manuscript lacked explicit confidence intervals and significance tests. In the revised version, we have updated the result tables to include Standard Deviations and Significance Testing.
> >
> > **6. Multi-Center Generalization**
> >
> > **Re:**
> > We appreciate the reviewer’s scrutiny regarding the heterogeneity of multi-center data. We addressed these variations through standardized preprocessing and validated the model's generalization via distinct internal and external test sets.
> >
> > Standardized Preprocessing to Control Variability: To mitigate domain shifts caused by varying acquisition protocols, we implemented a rigorous preprocessing pipeline. As shown in the table below, the raw data varies significantly in resolution and dimensions. We unified all images to a consistent voxel spacing of 0.34×0.34×0.55 mm^3 via spline interpolation and applied intensity normalization (percentile clipping and z-score) to standardize signal-to-noise ratios. This ensures the network learns robust anatomical features invariant to scanner-specific artifacts or contrast levels.
> >
> > Evidence of Multi-Center Generalization: We validated generalization using two distinct test sets: Set A (76 cases from public sources) and Set B (98 cases from an independent private center). Despite the wide range of acquisition parameters detailed in the table, our method achieves consistent high performance on both Set A and Set B. This stability across independent cohorts provides strong empirical evidence of the framework's robustness to multi-center variability.
> >
> > **Table: Statistics of Imaging Parameters in IAVS Dataset**
> > | Dataset Statistics | Min | Median | Max |
> > | :--- | :--- | :--- | :--- |
> > | Spacing (mm) | (0.21, 0.21, 0.30) | (0.36, 0.36, 0.50) | (0.47, 0.47, 1.20) |
> > | Volume Size (voxels) | (348, 384, 44) | (512, 512, 148) | (1024, 1024, 368) |

---

### Official Review · Reviewer_codZ · 2025-10-31

**Soundness:** 1
**Presentation:** 1
**Contribution:** 2
**Rating:** 2
**Confidence:** 4

**Summary:**

This paper aims to rethink the intracranial aneurysm vessel segmentation task from the perspective of computational fluid dynamics. This study has a dataset (new or enhanced), benchmark (new two metrics?), and a new method(?). As a reviewer, I believe the main issue with this paper is its excessive scope and the lack of a clear, focused contribution.

**Strengths:**

- This paper compiles and curates a large-scale 3D MRA dataset, combining existing datasets with a new in-house collection, resulting in a total of 641 volumes and IAs, which is quite impressive.
- The author's promise to open-source data, code, and models is great.

**Weaknesses:**

- The overall writing appears somewhat disorganized to the reviewer. If the author intends to claim a contribution to data construction, more detailed information is expected, such as how the data was constructed and which aneurysm-related tasks it supports. From the reviewer's perspective, proposing a new model is unnecessary, particularly since you are introducing new metrics simultaneously.
- In its current writing, the reviewer is unclear about how the existing dataset has been enhanced and why these enhancements are significant. Specifically, is CFD-based optimization necessary to improve the dataset, or can it simply be applied as a subsequent step after existing segmentation methods to support all aneurysm-related applications? The author should justify this.
- The loss function for stage 1 is strange. Why is the count classification loss needed?
- For the loss function of stage 2, please add explanations for each parameter....
- Figure 4 is difficult to understand. How does the author do "check vessel geometry"? Not using CFD?

**Questions:**

See questions in weakness.

Suggestions
- Rethinking the entire story of this paper is suggested. Or it is a huge waste of your dataset!

---

> ### Author Response · Authors · 2025-11-23
> **Response to Reviewer codZ Q1-Q3**
>
> We sincerely thank the reviewer for the critical and valuable feedback. We acknowledge that the initial presentation of our work suffered from a lack of focus regarding its core contribution. We have fundamentally restructured the paper to emphasize the **Benchmark Framework** as the primary contribution.
> We believe this new narrative significantly clarifies the value of our work. Below, we address your specific concerns point-by-point.
>
> **1. Scope and Contribution Clarity.**
>
> **Re:**
> We agree with the reviewer’s assessment. To address this, we have restructured the manuscript to position the **Benchmark Framework** as the core contribution.
> Although the dataset details were originally presented in the Appendix, we agree that this information is essential to understanding our benchmark. Therefore, in the revised version, we move key descriptions of dataset construction into the main text for better readability and completeness.
> We have also clarified that our proposed model serves primarily as a strong baseline for the benchmark and evaluation metrics, rather than as a new model contribution.
>
> **2. Dataset Enhancement & Necessity of CFD Optimization**
>
> **Re:**
> We appreciate the reviewer’s request for clarification. As shown in Table 1 ,existing public 3D MRA datasets provide only IA or IA-vessel masks, but lack the full set of geometric representations required for reliable downstream hemodynamic analysis. Our IAVS dataset is the first to provide all of these components at scale, and also the largest dataset.
> Importantly, CFD-based optimization is not merely a post-hoc step that can be applied after arbitrary segmentation outputs. Aneurysm CFD simulations are highly sensitive to geometric fidelity, including neck sharpness, lumen smoothness, and surface continuity. When segmentation results contain defects such as gaps at bifurcations, irregular lumen boundaries, or incomplete parent-vessel segments, it is often impossible to generate a valid mesh at all, preventing any downstream flow simulation. Without the standardized geometry preparation and quality control integrated in our pipeline, applying CFD as a separate post-processing step would therefore lead to solver failures or inconsistent hemodynamic fields.
> Rather than serving as an optional afterthought, the CFD-aware refinement is essential for transforming raw segmentations into simulation-ready geometries, and this is precisely the enhancement that distinguishes our dataset from existing ones. By providing CFD-valid geometries and consistent CFD outputs for 641 volumes and 587 aneurysms, our dataset supports both image-analysis research and clinically meaningful hemodynamic investigations, bridging a gap that previous datasets could not address.
>
> **3. Stage 1 Loss Function (Counting Classification)**
>
> **Re:**
> We thank the reviewers for their questions regarding the design of the loss function. The introduction of the counting classification loss aims to address the inherently high false positive rate in small aneurysm detection, a problem that was prominently observed during the development of our method.
>
> Pure heatmap regression focuses solely on local voxel-level responses. However, complex vascular structures often generate high-confidence local responses that resemble aneurysms, leading to numerous false positives when standard post-processing methods such as fixed thresholds or selecting the top-k candidates are applied.
>
> To mitigate this issue, we introduced the **counting classification loss** to impose global constraints. The heatmap loss guides the network to locate the precise positions of lesions based on local signals, while the counting loss forces the model to integrate features across the entire image patch to estimate the total number of aneurysms and suppress regions without aneurysms.
> These two components work together, the counting prediction acts as a global filter, helping to distinguish true lesions from vascular noise. During inference, this mechanism enables dynamic candidate selection. Instead of using a fixed number of candidates or a hard confidence threshold, we adaptively select the top $N$ clusters from the heatmap based on the predicted count N.
>
> As shown in our ablation study (**Table 3**), this strategy effectively suppresses background vascular noise, achieving the highest F1 score and recall rate compared to methods without the counting constraint.

---

> ### Author Response · Authors · 2025-11-23
> **Response to Reviewer codZ Q4-Q5**
>
> **4. Stage II Loss Function Parameters**
>
> **Response:**
> We appreciate the reviewer’s careful examination of our loss function formulation. We acknowledge that the definitions of specific parameters in Equation 2 were not explicitly detailed in the main text. We have added the following detailed explanations to the Methodology section and Appendix to clarify the mathematical notation.
>
> In Stage II, the total loss function is composed of a segmentation loss and a topology-aware clDice loss. The parameters are defined as follows:
>
> 1. The terms $p_i$ and $g_i$ represent the fundamental inputs for the loss calculation at each voxel $i$. Specifically, $p_i \in [0, 1]$ denotes the predicted probability output by the network indicating the likelihood of the voxel belonging to the vessel class. Correspondingly, $g_i \in {0, 1}$ represents the binary ground truth label, where 1 indicates the presence of a vessel and 0 indicates the background. The first term of the equation combines the Dice similarity coefficient and Cross-Entropy loss to supervise voxel-level segmentation accuracy.
>
> 2. The function $\mathcal{T}(\cdot)$ denotes the soft-skeletonization operation. Unlike standard morphological skeletonization which is non-differentiable, $\mathcal{T}(\cdot)$ employs an iterative "soft-skeletonization" algorithm consisting of min-pooling and max-pooling operations. This allows the extraction of the vascular centerline (skeleton) in a differentiable manner, enabling backpropagation during training. In our formula, $\mathcal{T}(p_i)$ represents the soft-skeleton extracted from the predicted segmentation, and $\mathcal{T}(g_i)$ represents the soft-skeleton derived from the ground truth mask.
>
> 3. The term involving $\mathcal{T}$ calculates the harmonic mean of precision and sensitivity between the predicted and ground truth skeletons. By maximizing the overlap between the extracted skeletons and the vascular masks, this term penalizes topological errors such as vessel breaks or false spurious connections that standard voxel-wise losses might miss.
>
> 4. The parameter $\lambda$ serves as a weighting factor that balances the contribution of the topological constraint against the standard segmentation accuracy. In our experiments, we empirically set $\lambda$ to ensure that the model prioritizes topological connectivity without compromising global volumetric overlap.
>
> We have updated the manuscript to include these formal definitions to ensure the reproducibility of our method.
>
> **5. Figure 4: Check Vessel Geometry**
>
> **Response:**
> We apologize for the confusion caused by the original figure. We have reorganized Figure 4 in the revised manuscript to enhance readability and clearly delineate the workflow logic.
>
> As illustrated in the updated figure, the evaluation system is explicitly divided into two major modules: "Check Vessel Geometry" and the "CFD Conversion Pipeline." To answer your specific question, the "Check Vessel Geometry" module is a topological verification step performed prior to the CFD pipeline to ensure the segmentation masks are geometrically valid for meshing. It does not involve CFD simulation but consists of two distinct steps:
>
> 1. Automated Topological Screening (marked in blue) : We compute Betti numbers to automatically detect gross topological errors, such as abnormal tunnels or cavities (holes) within the vessel structure.
>
> 2. Manual Adhesion Verification (marked in red): Since topological metrics cannot always distinguish between valid vascular connections and incorrect adhesions (where two adjacent vessels merge artificially), a manual review is conducted to identify and filter out these specific segmentation artifacts.
>
> Once the geometry is verified, it enters the CFD Conversion Pipeline, which is a fully automated process (marked in blue) that handles the end-to-end conversion from the mask to the simulation-ready files required for CFD analysis. The revised figure now uses color coding to clearly distinguish between these automated processes and the specific manual check.
>
> We once again thank the reviewer for the sharp and critical insights. Your feedback is instrumental in reshaping our manuscript. We believe that by centering the paper and clarifying the necessity of the CFD-aware dataset, the contribution is now distinct and substantial. We hope these revisions and clarifications address your concerns and demonstrate the value of our work to the community.

---

> > ### Comment · Reviewer_codZ · 2025-11-25
> > **Reply to rebuttal**
> >
> > Thank you for your reply. Your updated version is much clearer, which is great. However, I still find some parts confusing.
> >
> > 1. To facilitate the use of CFD models, you mention that "we establish two complementary benchmarks." However, since these are standard aneurysm tasks, it would be better to include some references and justify why you consider these two main usage scenarios. You cannot claim that you propose new things...
> >
> > 2. Therefore, I find it unusual to modify the Stage 1 loss function. Typically, localization methods include segmentation-based localization, bounding box detection, and heatmap regression, which are well-established techniques in aneurysm localization. Could you provide a rationale for introducing a new heatmap regression function? Are these baseline methods using the same loss function?
> >
> > 3. But I understand that introducing a new loss function for Stage 2 is important because it incorporates topology information.
> >
> > 4. I recommend reorganizing Sections 4 and 5. I think your dataset does not contribute to the localization task. Based on your loss function, modifying this can not be inspired by your new dataset construction. Since it is effective, it could serve as the basis for another, separate paper. Conversely, for segmentation, including topological connectivity can significantly improve performance. Ultimately, attempting to apply your dataset to all aneurysm tasks may not be necessary; focusing on tasks like segmentation, which truly benefit from this additional information, should be sufficient and highlight your dataset contribution.

---

> > > ### Author Response · Authors · 2025-11-28
> > > **Reply to Response of Reviewer codZ**
> > >
> > > We appreciate your continued thoughtful feedback. We address your points as follows:
> > >
> > > **R1:**
> > >
> > > We agree that global localization and local segmentation tasks are well-established in medical image analysis. Our motivation for establishing these two benchmarks is based on the realization that the integration of CFD simulations into the pipeline requires standardized, high-fidelity geometries that are not typically available in existing benchmarks. We will revise the manuscript to provide more detailed references to similar works in the aneurysm tasks.
> > >
> > > **R2:**
> > >
> > > For the evaluation of Stage 1, we follow the setting of [1], where a predicted point is considered correct if its distance to the ground truth is smaller than the aneurysm diameter. This criterion allows for minor spatial deviations, which is acceptable because the downstream segmentation stage does not require pixel-level precision in center localization. Under this setting, the more critical optimization objective becomes the reduction of false positives rather than further improving sub-voxel localization accuracy. In practice, false positives directly trigger incorrect vascular segment segmentation and severely degrade the final CFD analysis results, making them far more harmful than small localization errors.
> > >
> > > Although heatmap regression itself is a standard technique in keypoint detection, our modification does not aim to introduce a new general localization theory. Instead, to explicitly address the false-positive issue, we design a loss function that combines a heatmap regression loss and a count classification loss. The heatmap loss follows the widely adopted Focal Loss formulation used in keypoint detection [2], with only minor empirical adjustments (e.g., setting the positive voxel threshold to 0.9), which are validated to be effective for this task. More importantly, the count classification loss introduces a global constraint on the predicted number of aneurysms, thereby suppressing spurious detections by leveraging global image features. This coupling allows the network to better distinguish true aneurysm candidates from background vascular noise, particularly benefiting small aneurysm detection.
> > >
> > > Compared with segmentation-based localization and bounding-box detection, our design is an engineering optimization tailored to this specific task to substantially reduce false positives and establish a strong and reliable baseline for cerebral aneurysm localization in support of downstream segmentation and CFD analysis.
> > >
> > > **R3:**
> > >
> > > We appreciate your understanding of the importance of the Stage 2 loss function.
> > >
> > > **R4:**
> > >
> > > We fully agree with your viewpoint that the core innovation and primary value of the IAVS dataset lie in segmentation and CFD-valid geometry construction rather than in localization. As clarified above, our dataset is fundamentally designed to support high-fidelity aneurysm and vessel segmentation for downstream hemodynamic analysis.
> > >
> > > Regarding Stage I, we would like to clarify that the motivation for introducing the counting-guided heatmap formulation is not to claim a novel method, but an engineering optimization tailored to this specific task, aiming to substantially reduce false positives by constraining the predicted count and thereby establishing a strong baseline for cerebral aneurysm localization.
> > > While detection itself is not the core contribution of our dataset, improving the reliability of candidate localization directly benefits the end-to-end segmentation performance, as it provides cleaner and more accurate region proposals for Stage II. Consequently, this also improves the efficiency and robustness of the downstream CFD pipeline by reducing unnecessary or erroneous segmentation attempts.
> > >
> > > Following your suggestion, we will Sections 4 and 5 to reposition the main contribution clearly. Once again, we sincerely thank you for this insightful suggestion, which has helped us significantly sharpen the scope and presentation of our contributions.
> > >
> > > [1] Zi-Hao Bo. et al., Toward human intervention-free clinical diagnosis of intracranial aneurysm via deep neural network. Patterns, 2021.
> > >
> > > [2] Zhou, Xingyi. et al. Objects as points. Computer Vision and Pattern Recognition, 2019.

---

### Official Review · Reviewer_bKFQ · 2025-11-08

**Soundness:** 3
**Presentation:** 3
**Contribution:** 3
**Rating:** 8
**Confidence:** 5

**Summary:**

This article proposes to tackle the problem of vessel segmentation associated with haemodynamic analysis using Computational Fluid Dynamics (CFD). In particular, this article proposes to study intracranial aneurysms detection. The reason why CFD is involved in this study is to help clinician assess the rupture risk of aneurisms.

Authors propose a novel multi-center dataset that includes vessel masks, centreline, meshing and CFD results; two evaluation benchmarks, one for the localisation of aneurisms and the second for the accurate segmentation of blood vessels. Further, the authors make the point that good voxel-wise segmentation results do not guarantee something that will mesh correctly or results in accurate CFD simulations. Consequently they also propose a CFD suitability score system.

The paper describes their proposal and evaluation for vessel and aneurism detection and segmentation, the preprocessing steps leading to meshing and CFD simulation in the vessel region around detected aneurisms.

**Strengths:**

The paper describes a novel attempt at setting up a complete system of intracranial aneurism detection and evaluation based on 3D MR angiography. The paper is very interesting, well presented and although not perfect, quite sound in its approach. Given the amount of work necessary to collect, annotate and perform CFD simulation on more than 600 MRA volumes, the proposed dataset is quite unique and valuable.

The evaluation results are clearly the current state of the art among academic published papers although there exist proprietary datasets and methods in the same broad research domain. The availability of code and models will certainly advance the state of the art in this domain.

**Weaknesses:**

Not very many articles tackles the joint segmentation and CFD aspects of vessel segmentation, yet this has been an ongoing research area for a quite time. The article focuses on relatively recent, deep-learning based methods and does not attempt to review previous classical efforts. Previous to deep-learning, vessel segmentation methods used vesselness methods [1], many of which were reviewed and evaluated in [2]. Relatively recent projects have contributed to very similar objectives [3].

Overall the paper reads well but some important details are obscured in supplementary material, for example the CFD readiness evaluation is only partly automated. Difficult cases still need to be corrected by hand. This is not clear in the main article.

It is not absolutely clear what CFD results present in the proposed dataset consist off, and what they are used for. In the paper they are not evaluated, only the amenability of result for CFD computation is. It would be extremely useful to associate the CFD results with some clinical evaluation, e.g is an aneurism dangerous? does it need to be operated on? What are the criteria, etc. Since this is the claimed end result of the proposed pipeline, this would seem essential.

The vessels segmentation method uses the clDice method which does not offer guarantee of connectivity contrary to what the authors imply. Indeed the skeleton the clDice is the medial axis, computed through the Lantuejoul formula, which does not yield connected voxels. Instead they should use a truly connected skeleton computable by deep-learning layers [4] or the skeleton recall loss [5] which is connected and very fast. Their results would likely improve.



[1] A.F. Frangi, W.J. Niessen, L.V. Koen, and M.A. Viergever. Multiscale vessel enhancement filtering.
Lecture Notes in Computer Science, 1496:130ff., 1998.
[2] Jonas Lamy, Odyssee Merveille, Bertrand Kerautret, and Nicolas Passat. A benchmark framework for
multiregion analysis of vesselness filters. IEEE Transactions on Medical Imaging, 41(12):3649–3662, 2022.
[3] https://explore.openaire.eu/search/project?projectId=anr_________::8e389f59b7c7aa1d24104847c23b5b09
[4] Mario Viti, Hugues Talbot, Bassam Abdallah, Etienne Perot, and Nicolas Gogin. Coronary artery
centerline tracking with the morphological skeleton loss. In 2022 IEEE International Conference on
Image Processing (ICIP), pages 2741–2745. IEEE, 2022.
[5] Yannick Kirchhoff, Maximilian R Rokuss, Saikat Roy, Balint Kovacs, Constantin Ulrich, Tassilo Wald,
Maximilian Zenk, Philipp Vollmuth, Jens Kleesiek, Fabian Isensee, et al. Skeleton recall loss for connec-
tivity conserving and resource efficient segmentation of thin tubular structures. In European Conference
on Computer Vision, pages 218–234. Springer, 2024

**Questions:**

- Why not perform CFD through the entire segmented vessel network?
- Blood is not a Newtonian fluid, why choose a Newtonian CFD solver?

**Details Of Ethics Concerns:**

The authors propose a highly biometric dataset (human MR angiography, segmented intra-cranial blood vessels). It is not clear if their data is GDRP-compliant. The authors should make that statement.

---

> ### Author Response · Authors · 2025-11-22
> **Response to Reviewer bKFQ Q1-Q4**
>
> We thank the reviewer for the positive assessment of our paper’s readability and the constructive feedback. We appreciate the detailed suggestions regarding the literature review, connectivity losses, and CFD validation. We have carefully addressed each point below and revised the manuscript accordingly.
>
> **1. Literature Review & Classical Methods**: The article focuses on deep-learning methods and misses classical vesselness methods. Suggestions to include references [1], [2], and [3].
>
> **Re:**
> We appreciate the valuable references provided. Following the suggestion, we have updated the Related Work section to include a discussion of these classical approaches.
>
> **2. Automation of CFD Applicability**: The CFD readiness evaluation is only partly automated, and difficult cases need hand correction. This is not clear in the main article.
>
> **Re:**
> We agree that the degree of automation in the CFD-readiness evaluation was not sufficiently emphasized. Although some manual steps are required for challenging anatomies, **the majority of the pipeline’s processes are fully automated**. In the revised version of Figure 4, we have more clearly delineated the automated and manual components to enhance transparency regarding the pipeline’s current capabilities. We have clarified this distinction in the Methodology section of the revised manuscript to ensure transparency regarding the pipeline's current capabilities.
>
> **3. Clinical Relevance of CFD Results**: It is not clear what the CFD results are used for or their clinical evaluation.
>
> **Re:**
> Thank you for pointing this out. We acknowledge that the manuscript did not clearly describe the clinical relevance of the CFD outputs in our dataset. We agree that the ultimate value of CFD lies in its ability to inform clinical decision-making, such as assessing rupture risk or surgical urgency. To clarify, our simulations generate the foundational **hemodynamic biomarkers**, specifically Wall Shear Stress (WSS), intra-aneurysmal pressure, and velocity fields—that are standardly used to evaluate aneurysm danger levels.
> In clinical contexts, these are not merely physical quantities but critical diagnostic indicators. For instance, **WSS** is a primary predictor for aneurysm remodeling, where low WSS regions are associated with flow stagnation and inflammatory cell infiltration (leading to wall degeneration), while high WSS correlates with initiation and potential rupture points. Similarly, **pressure distribution** serves as a direct proxy for wall tension and mechanical load, and **velocity fields** help identify flow impingement zones that accelerate wall thinning. Therefore, the CFD outputs we compute correspond exactly to the quantities that downstream clinical studies analyze. While our paper technically focuses on the pipeline's fidelity, verifying the CFD-amenability of the geometries is a necessary prerequisite for diagnosis, as poor segmentation would lead to erroneous risk biomarker calculations. We have revised the Introduction and Discussion to explicitly map these CFD outputs to their specific clinical diagnostic criteria (e.g., rupture risk and growth prediction) to clarify this connection.
>
>
> **4. Connectivity and Topological Loss Functions**: The clDice method does not guarantee connectivity. Suggestions to use a truly connected skeleton [4] or Skeleton Recall Loss [5].
>
> **Response:**
> We thank the reviewer for this deep technical insight. We acknowledge that clDice relies on a soft skeleton approximation. We took the suggestion seriously and performed additional experiments to compare our baseline against the **Skeleton Recall Loss** [5].
> Our empirical results indicate that clDice achieves better performance for our specific task. As shown in the table below, the model trained with clDice loss outperformed Skeleton Recall Loss on both Dice and clDice metrics. This suggests that for the specific geometry of intracranial aneurysms, the soft constraint provided by clDice offers a more effective gradient signal. Regarding reference [4], we were unable to conduct a direct comparison as the source code is not publicly available.
>
> We have retained clDice as a strong baseline but added a discussion in the revised manuscript highlighting [4] and [5] as potential directions for future algorithmic improvements.
>
> | Method | Loss | Set A-Dice | Set A-HD95 | Set A-clDice | Set A-BIoU | Set B-Dice | Set B-HD95 | Set B-clDice | Set B-BIoU |
> | :--- | :--- | :--- | :--- | :--- | :--- |:--- | :--- | :--- | :--- |
> | nnUNet | clDice Loss | **0.8563** | **3.2809** | **0.8629** | **0.7576** | **0.8368** | 4.2134 | **0.8616** | **0.7388** |
> | nnUNet | Skeleton Recall Loss | 0.8401 | 3.5820 | 0.8447 | 0.7350 | 0.8296 | **4.1835** | 0.8516 | 0.7303 |

---

> > ### Author Response · Authors · 2025-11-22
> > **Response to Reviewer bKFQ Q5**
> >
> > **5. CFD Simulation**: Why not perform CFD through the entire segmented vessel network?
> >
> > **Re:**
> > We thank the reviewer for their insightful comment. We would like to clarify that our decision to focus segmentation and CFD simulation on the aneurysm and its adjacent parent vessels, rather than the entire vascular network, was a deliberate and principled choice. This approach aligns with established standards in the field and is based on a well-considered trade-off between clinical relevance and computational feasibility.
> >
> > Firstly, in the field of 3D aneurysm segmentation from medical images, it is standard practice to focus on the local geometry (i.e., the aneurysm and its parent vessel) [1, 2]. This approach is favored because it captures the clinically critical structures required for hemodynamic analysis. As noted in a recent comprehensive survey by Hsu et al. [3], this local focus is the mainstream methodology. This practice not only simplifies model training but, more importantly, ensures the applicability of the segmentation for downstream CFD simulation. Our IAVS dataset is designed in accordance with this established standard.
> >
> > Secondly, the 3D models for CFD are derived directly from these segmentations. We opted for a local model because aneurysm rupture risk assessment is primarily dependent on local hemodynamic parameters—such as pressure, velocity, and Wall Shear Stress (WSS)—rather than global flow dynamics of the entire network. The validity and sufficiency of using local vessel models for aneurysm analysis have been demonstrated and validated in numerous studies [4-6].
> >
> > Furthermore, simulating the entire cerebrovascular network, which can involve thousands of arterial branches, is computationally prohibitive. Such simulations often require weeks of supercomputer resources, rendering them infeasible for large-scale datasets or clinical translation. In contrast, our local simulations can be completed within days. This significant efficiency gain is critical for practical application and avoids unnecessary computational overhead. Our IAVS dataset and evaluation pipeline are standardized on this local, efficient approach, which has proven sufficient for risk-related hemodynamic analysis.
> >
> > In conclusion, our method is not an evasion of complexity but a pragmatic decision to prioritize the utility, accuracy, and feasibility of local simulation. By developing CFD-guided evaluation systems (such as our clDice loss and CFD-AS metrics) for this local region, we align with the field's consensus: local CFD analysis is sufficient for revealing the key hemodynamic mechanisms. This focused approach provides a robust and feasible foundation for future end-to-end optimization, such as the integration of physics-informed losses.
> >
> > We will update the manuscript to explicitly articulate this rationale, citing relevant literature to support the sufficiency of local models for risk assessment.
> >
> > **References**
> >
> > [1] Yang X. et al., INTRA: 3D intracranial aneurysm dataset for deep learning, CVPR, 2020.
> >
> > [2] Zhang J. et al., Edge-oriented Point-cloud Transformer for 3D Intracranial Aneurysm Segmentation, Proc. MICCAI 2022, 2022.
> >
> > [3] Hsu W.C. et al., A survey of intracranial aneurysm detection and segmentation, Medical Image Analysis, 2025.
> >
> > [4] Song M. et al., Intracranial aneurysm CTA images and 3D models dataset with clinical morphological and hemodynamic data, Scientific Data, 2024.
> >
> > [5] Kliś K.M. et al., Can $\beta$-blockers prevent intracranial aneurysm rupture?: insights from Computational Fluid Dynamics analysis, Cardiovascular Research, 2024.
> >
> > [6] Souza A. et al., Experimental and numerical analyses of the hemodynamics impact on real intracranial aneurysms: A particle tracking approach, Results in Engineering, 2024.

---

> > > ### Author Response · Authors · 2025-11-22
> > > **Response to Reviewer bKFQ Q6-Q7**
> > >
> > > **6.  Blood is not a Newtonian fluid, why choose a Newtonian CFD solver?**
> > >
> > > **Re:**
> > > We thank the reviewer for raising this important question. This highlights a classic trade-off in the CFD simulation of intracranial aneurysms. Our choice of a Newtonian fluid model (implemented in OpenFOAM) was not an oversight of blood's non-Newtonian properties. Instead, it was based on a thorough analysis of the relevant physiological context and the broad consensus in the research community. For the specific application of intracranial aneurysms, approximating blood as a Newtonian fluid is a validated and widely accepted standard practice.
> > > Blood's non-Newtonian (shear-thinning) characteristics are primarily significant in low shear rate regions, such as the microcirculation in capillaries (<0.5 mm). Our region of interest involves large intracranial arteries, including the aneurysm and its parent vessels. In these areas, blood flow is relatively high, generating high shear rates (typically > 100 $s^{-1}$). Under this high shear rate regime, the apparent viscosity of blood approaches a constant plateau (approximately 0.0035-0.4 Pa·s). Its rheological behavior becomes highly consistent with that of a Newtonian model, which assumes a constant viscosity [1].
> > > Based on this, numerous comparative studies, including the analysis in [2], have confirmed that for intracranial aneurysm simulation, the differences in key hemodynamic parameters (like WSS distribution) between Newtonian and non-Newtonian models are minimal (typically < 5-10%). Importantly, the spatial distribution patterns from both models are consistent and do not alter the final clinical risk stratification conclusions. The study in [2] also supports that the Newtonian approximation is sufficient for unruptured aneurysms. Consequently, as noted in the survey by [2], the vast majority (approximately **90%**) of CFD studies for initial risk screening employ the Newtonian model, recognizing it as standard practice. Non-Newtonian models are generally considered necessary only for specific, fine-grained analyses, such as evaluating local wall effects after stent placement.
> > > In summary, our choice of a Newtonian CFD model was a deliberate decision grounded in strong physiological evidence and widespread community consensus. This approach ensures the accuracy and reliability of our results while also guaranteeing the practical utility of our study, allowing for direct comparison with the vast majority of existing work in this field.
> > >
> > > **References**
> > >
> > > [1] Alnæs M.S. et al., Computation of hemodynamics in the circle of Willis, Stroke, 2007.
> > >
> > > [2] Huang F. et al., On flow fluctuations in ruptured and unruptured intracranial aneurysms: resolved numerical study, Scientific Reports, 2024.
> > >
> > > **7. Data Privacy**
> > >
> > > **Re:**
> > > Thank you for raising this important point. Our dataset is constructed through a combination of publicly available MRA datasets and institutional clinical data collected under approved protocols.

---

### Author Response · Authors · 2025-12-04
**Summary of Rebuttal and Response to Reviewers**

Dear Area Chair:

We present IAVS, the first large-scale (641 volumes) benchmark bridging medical segmentation and Computational Fluid Dynamics (CFD). While we received a Strong Accept (**Score 8**) recognizing the work as "unique and valuable," we have also resolved the concerns of other reviewers through manuscript restructuring and extensive new experiments.

**Response to Reviewer bKFQ (Score: 8):**

We thank the reviewer for recognizing our dataset as "quite unique" and our results as state-of-the-art. We agree that linking segmentation to hemodynamics is vital. We have revised the text to explicitly map CFD outputs (WSS, pressure) to clinical diagnostic criteria, clarifying the practical value of our pipeline.

Regarding the suggestion to use Skeleton Recall Loss, we conducted a comparative experiment. Results show **clDice consistently outperforms Skeleton Recall Loss** on our dataset (Dice: 0.856 vs. 0.840). We thus retained clDice but highlighted the suggested method as a valuable future direction.

**Response to Reviewer codZ (Score: 2):**

Crucially, the initial low score stemmed from a misunderstanding of the paper’s scope. After we restructured the manuscript to emphasize the Benchmark Framework, the reviewer acknowledged in the discussion: **"Your updated version is much clearer, which is great."** This positive shift indicates the fundamental concerns regarding contribution have been resolved.

Technically, we clarified that the **Stage 1 counting loss** is a specific engineering optimization to suppress false positives. We also provided the detailed mathematical formulation for Stage 2 parameters. We believe the initial "Reject" rating no longer reflects the current clarity of the revised manuscript.

**Response to Reviewer j58o (Score: 4):**

To address concerns about topological weights compromising accuracy, we conducted a **sensitivity analysis** on loss weight (α). Results confirm that a balanced weight (α=0.5) enhances connectivity without distorting local geometry, actually improving Dice (0.853→0.856). This resolves the concern about geometric sacrifice.

Regarding CFD soundness, we added rigorous validation details: a **grid independence study** (convergence at 0.10mm mesh with <0.25% WSS difference) and stability criteria (CFL < 1, residuals < 10^−4). This proves our CFD-Applicability Score is a scientifically grounded metric backed by numerical stability.

**Response to Reviewer oRdZ (Score: 4):**

Regarding "methodological novelty," we emphasize that as a Benchmark contribution, our value lies in the standardized pipeline and CFD metrics. However, to strengthen baselines, we compared against the recent SOTA **Zig-RiR (IEEE TMI 2025)**. Our optimized nnU-Net significantly outperformed it (Dice: 0.856 vs. 0.706), validating our framework's effectiveness.

Regarding "External Validation," we demonstrated robustness across our multi-center data: **Set A (Public) vs. Set B (Private)**. Despite different scanners and protocols, our model achieved consistent high performance on both, providing strong empirical evidence of generalization.

**Conclusion**

We have actively addressed all concerns. With **Reviewer codZ admitting the paper is "much clearer"** and the **strong endorsement from Reviewer bKFQ**, we believe the IAVS benchmark is ready to serve the community. We respectfully ask the Area Chair to consider these substantial improvements.

---

### Meta-Review · Area_Chair_8Co1 · 2025-12-16

**Summary:**

This work introduces the Intracranial Aneurysm Vessel Segmentation (IAVS) dataset, comprising 641 3D MRA images with annotations of aneurysms and IA-Vessels. Four reviewers have reviewed this paper, with one acceptance (rating 8), two marginally below the acceptance threshold (rating 4), and one rejection (rating 2). The overall rating is negative and below the acceptance threshold.

First, the authors did not respond to the Ethics Review of the Area chair. I found that the papers said that the dataset has been approved by the IRB, but the details were not given. I guess it is due to the anonymous policy of ICLR.

Reviewer bKFQ is inclined to the positive aspect, and the authors have carefully addressed the raised concerns. However, Reviewer bKFQ asked *“if their data is GDRP-compliant. The authors should make that statement”*. It is weird that the authors did not answer this question.

Reviewer codZ is concerned about the descriptions, the datasets' details, implementation, and especially the contributions (*“lack of a clear, focused contribution”*). **First of all, I have to say the authors' overall summary is misleading and dishonest**, i.e., “the reviewer acknowledged in the discussion … the fundamental concerns regarding contribution have been resolved.” Actually, Reviewer codZ is still concerned about the method and contributions.

Reviewer j58o is mainly concerned about the experiments, and the authors have added extensive experiments and analysis in the rebuttal. Most of the concerns have been well addressed. But I think the authors did not adequately address the multi-center generalization capability, since only two cohorts are evaluated. As highlighted in the abstract, IAVS is a *“large-scale multi-center dataset”*, I think only 76 and 98 cases from two centers may not be enough. It is a potential over-claim.

Reviewer oRdZ is mainly concerned about the technical contribution and points out the lack of deeper learning or modeling insights, while the authors agreed reviewer’s opinion. I appreciate the honest answer of the authors. For other parts, I think the authors answered well, with additional experiments and discussions.

Overall: Although this work shows strengths and contributions to a new dataset, I am sorry to recommend rejection due to unresolved concerns, negative ratings, and the lack of technical contributions.

**Reviewer Concerns:**

Reviewers bKFQ may keep the positive rating. Some concerns of reviewers codZ and oRdZ are not addressed.

**Reviewer Scores:**

Reviewer j58o may slightly improve the rating to 6.

---

### Decision · Program_Chairs · 2026-01-26

Reject